



# ACDL/DQ-1 Calibration Algorithms. Part I: Nighttime 532 nm Polarization and High-Spectral-Resolution Channel

Fanqian Meng[1], Junwu Tang[2], Guangyao Dai[1], Wenrui Long[1], Kangwen Sun[1], Zhiyu Zhang[1], Xiaoquan Song[1,2], Jiqiao Liu[3], Weibiao Chen[3], Songhua Wu[1,2,4]

[1]College of Marine Technology, Faculty of Information Science and Engineering, Ocean University of China, Qingdao, 266100, China
[2]Laoshan Laboratory, Qingdao, 266200, China
[3]Key Laboratory of Space Laser Communication and Detection Technology, Shanghai Institute of Optics and Fine Mechanics, Chinese Academy of Sciences, Shanghai 201800, China
[4]Institute for Advanced Ocean Study, Ocean University of China, Qingdao, 266100, China

*Correspondence to*: Guangyao Dai, daiguangyao@ouc.edu.cn

**Abstract.** The Atmospheric Environment Monitoring Satellite (DQ-1) was successfully launched in April 2022, with the capability of providing continuous multi-sensor spatial and optical simultaneous observations of carbon dioxide, aerosols and clouds. The primary payload carried on DQ-1 is an Aerosol and Carbon dioxide Detection Lidar (ACDL). The instrument comprises a high-spectral-resolution channel at 532 nm, elastic channels at 532 nm and 1064 nm, and integrated-path differential absorption (IPDA) channel at 1572 nm. The optical properties of aerosols and clouds measured by the ACDL promote a quantitative characterization of the uncertainties in the global climate system, hence the precise calibrations for the ACDL are necessary. This paper outlines the algorithms employed for calibrating the nighttime 532 nm measurements for the first spaceborne high-spectral-resolution lidar with an iodine vapor absorption filter. The nighttime calibrations of the 532 nm data are fundamental to the ACDL measurement procedure, as they are utilized to derive the calibrations over daytime orbits and the calibrations of the 1064 nm channel relative to the 532 nm channel. This paper provides a review of the theoretical foundations for molecular normalization techniques as applied to spaceborne lidar measurements, includes a detailed discussion of auxiliary data and theoretical parameters used in ACDL calibrations, as well as a comprehensive description of the calibration algorithm procedure. To mitigate large errors stemming from high-energy events during calibration, a data filter is designed to obtain valid calibration signals. The paper also assesses the results of the calibration procedure, by analysing the errors of calibration coefficients and validating the attenuated backscatter coefficient results. The results indicate that the relative error of the calibrated attenuated backscatter coefficients is lower than 2% in the calibration area, and the uncertainty of the pure-molecule attenuated scattering ratio was within anticipated range of 5%.



## 1 Introduction


The Atmospheric Environment Monitoring Satellite (DQ-1), which was launched on April 16, 2022, is a research satellite designed to monitor the atmospheric environment. It is equipped with five payloads (Zhu et al., 2023) including an Aerosol and Carbon Detection Lidar (ACDL), a Particulate Observing Scanning Polarimeter (POSP), a Directional Polarization Camera (DPC), an Environmental Trace Gases Monitoring Instrument (EMI), and a Wide Swath Imager (WSI). The primary payload

is the ACDL, which is a lidar system consisting of two different modules. One is the aerosol-measurement module which provides aerosols and clouds profile measurements with high accuracy globally, and another is the $CO_2$ measurement module for atmospheric column $CO_2$ observations (Liu et al., 2019; Wang et al., 2020). The scientific objective of the ACDL is to detect high-resolution vertical profiles of global atmospheric aerosols and clouds. It aims to explore the optical features of atmospheric aerosols and clouds, and gather information related to the distribution of global atmospheric column $CO_2$

concentrations, provide precise quantitative scientific data for determining the sources and sinks of $CO_2$ (Chen et al., 2023).

To enable ACDL quantitative measurement of atmospheric parameters, the calibration for the raw measurement data is necessary. The spaceborne lidar signal comprises lidar specifications, measured distance, particle backscatter signal and atmospheric attenuation. The calibration procedure for the spaceborne lidar is defined as the construction of a quantitative relationship between the particle backscatter signal and the corresponding lidar signal. The calibration procedure calculates

the calibration coefficients for each channel and applies the calibration coefficients to the original profiles to obtain the attenuated backscatter coefficients. The nighttime 532 nm polarization and high-spectral resolution channels (hereafter referred to as high-spectral-resolution channel) of the ACDL were calibrated utilizing molecular normalization calibration techniques. The calibrations were conducted in areas with clean atmosphere, where aerosols and clouds are absent, all backscattered light is assumed to be of molecular origin. The accurate estimation of expected backscatter is calculated from the European Centre

for Medium-Range Weather Forecasts (ECMWF) atmospheric assimilation model of the fifth generation reanalysis (ERA5) dataset (Hersbach et al., 2020). The resulting calibrated attenuation backscatter coefficient product serves as a foundation for the subsequent lidar products, with accurate calibration results being crucial for ensuring the credibility of those products.

The currently operational spaceborne lidars have formulated calibration algorithms based on the molecular normalization calibration technique specific to their own characteristics, and conducted calibrations in clean atmospheric regions. The Lidar

In-space Technology Experiment (LITE) lidar system uses data at the height of 30 and 34 km to derive calibration coefficients for the 355 nm and 532 nm channels, with calibration coefficients maintained at a range of ±5% (Osborn, 1998; Ressell et al., 1979). Based on the experiences of the LITE, the Cloud-Aerosol Lidar and Infrared Pathfinder Satellite Observations (CALIPSO) employs a calibration approach using molecular normalization technique applicable to the 532 nm, and cirrus spectral backscatter ratio for the 1064 nm channels of the Cloud-Aerosol Lidar with Orthogonal Polarization (CALIOP, one

of instruments aboard the CALIPSO). The calibration of the CALIOP undergone four releases, in the first three releases, the atmosphere was used as the calibration altitude at 30– 34 km consistent with LITE (Russell et al., 1979; Reagan et al., 2002; Hostettler et al., 2006; Powell et al., 2009). Additionally, the CALIPSO scientific team has formulated calibration algorithms



for the 532 nm daytime orbit and the 1064 nm channel (Powell et al., 2010; Vaughan et al., 2010). However, the studies have shown that aerosols within the 30–34 km exhibit temporal and spatial variabilities, indicating that they cannot be disregarded

(Vernier et al., 2009). Therefore, CALIPSO updated the stratospheric molecular normalization region up to 36–39 km in a subsequent revision and adjusted the corresponding algorithms (Kar et al., 2018). Additionally, the calibration algorithms for the 532 nm daytime and 1064 nm channels were also revised (Getzewich et al., 2018; Vaughan et al., 2019). The Cloud-Aerosol Transport System (CATS) on the space station is designed for detecting cloud and aerosol. Since its calibration region is selected between 23 and 27 km, it cannot fully disregard aerosol effects and the system also considers the impact of the

aerosol stratospheric scattering ratio (Yorks et al., 2016). Compared to cloud and aerosol detection lidars like CALIPSO, the the Ice, Cloud, and land Elevation Satellite (ICESat) series of satellites, primarily focus on measuring the elevation of ice sheets, glaciers, sea ice and more. They are calibrated using molecular normalization technique as well. The calibration altitude range for the Geoscience Laser Altimeter System (GLAS) on ICESat was selected as 26–30 km (Palm et al., 2011), whereas the Advanced Topographic Laser Altimeter System (ATLAS) lidar system on its improved instrument ICESat-2 with selected

chose the region of 11–13.5 km altitude in the 60–65 degree polar for calibration due to data frame limitations (Palm et al., 2021). Furthermore, the Atmospheric LAser Doppler INstrument (ALADIN) onboard ADM-Aeolus (Atmospheric Dynamics Mission-Aeolus), was a direct detection Doppler Wind Lidar operated in the ultra-violet region, also performed the calibration of attenuated backscatter coefficient. The ALADIN sets the calibration in the atmospheric altitude range of 6–16 km at mid to high latitudes. The calibration coefficients for Rayleigh and Mie scattering channels are also calibrated using the molecular

normalized technique (Pierre et al., 2020). The upcoming deployment of The Earth Cloud, Aerosol and Radiation Explorer (Earth-CARE) lidar system has nearly finished ground-based calibration and performance verifications, with post-launch on-orbit calibrations scheduled to follow (Wehr et al., 2023).

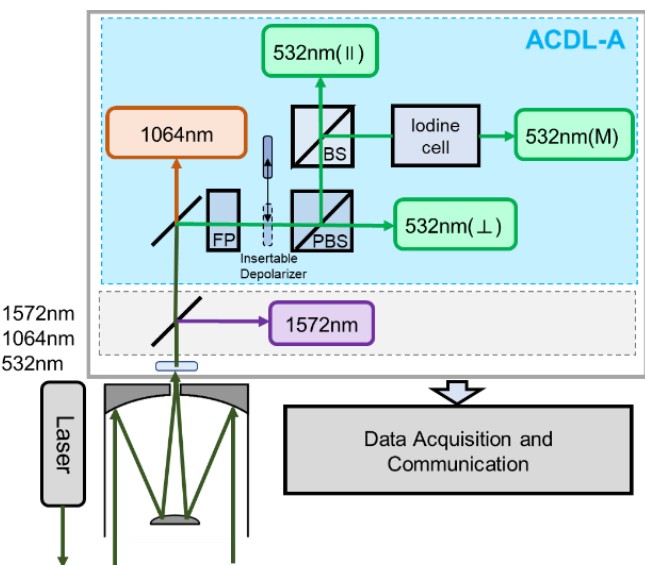

**Figure 1: Schematic diagram of ACDL (duplication from Dai et al., 2023).**



The ACDL receiver subsystem gathers the echoed signal through five channels: three channels at 532 nm, one channel at 1064 nm, and one channel at 1572 nm, as illustrated in Figure 1. After passing through the polarization beam splitter (PBS), the 532 nm signals are split into cross-polarized and parallel-polarized components separately. The entire parallel-polarized signal passes through a beam splitter (BS), while a portion (70%) of the parallel–polarized signal passes through an iodine vapor absorption filter to block Mie scattering, thus constituting the high-spectral-resolution channel. The remaining signal enters

the parallel-polarization channel. The 1064 nm channel maintains the same sampling frequency as the 532 nm channels to detect atmospheric conditions, and the 1572 nm channel identifies $CO_2$ concentrations. The backscattered photons then excite the photomultiplier tube (PMT) located in each channel, which converts light into electrical signals. Calibration procedure converts electrical signals to backscatter coefficients for calculating atmospheric and aerosol products. The ACDL scientific team has initially achieved total depolarization ratio, backscatter coefficient, extinction coefficient, lidar ratio, color ratio, and

other optical parameter products of aerosols and clouds. These products provide a characterization of the rich hierarchical structure of global aerosols and clouds in both vertical and horizontal directions (Dai et al., 2023).

This paper outlines the calibration methodology for the ACDL 532 nm polarization and high spectral resolution channels, shows the results of the global calibration coefficients and attenuated backscatter coefficients, and assesses the results. Section 2 describes the calibration algorithms for the ACDL. Section 3 highlights the corresponding validation methods applied.

Discussions, conclusions and outlook are summarized in Sections 4 and 5.

## 2. Nighttime Calibration Algorithms and methodology

Calibration procedure is a fundamental element of processing spaceborne lidar data, and its goal is to establish a quantitative relationship between the particle backscatter coefficient and the electrical signals detected by the lidar system. The ACDL 532 nm channel comprises three channels that receive parallel-polarized signal, cross-polarized signal and high-spectral-resolution

signal. The calibration procedure is based on the original range-scaled energy and gain-normalized signal (hereinafter normalized signal). It requires system parameters, including signal distance, pulse energy, gain and so on. Among these, the output pulse energy of the ACDL laser pulse is measured by an energy monitor. Table 1 lists the lidar parameters utilized for the calibration.

**Table 1: Parameters of the ACDL instrument for calibration**

| Parameters | Value |
|---|---|
| Wavelength | 532.024 nm; |
| Pulse Energy | ~130 mJ@532 nm; |
| Gain | 59.46@parallel; 53.4573@vertical; 32@ high-spectral-resolution |
| Lidar Off-Nadir Angle | 2° |
| Laser Repetition Frequency | 40Hz@532nm |
| Vertical Resolution | 3 m@<7.5 km; 24 m (8 bin average) @>7.5 km |
| Horizontal Resolution | ~ 330 m |




Both the nighttime parallel and high-spectral-resolution channels of ACDL were calibrated using the molecular normalization calibration technique, consists of the following steps:

Step 1: The molecular transmittance and ozone absorption at 532 nm within the calibration regions were computed using the ERA5 atmospheric prediction model data provided by ECMWF. Additionally, the molecular backscattering at the corresponding location was calculated based on its pressure and temperature data. Then the transmittance effects due to the

Fabry-Pérot etalon (F-P etalon) and the iodine vapor absorption filter following the ACDL system design are computed;

Step 2: Match the denoised lidar signal with the above calculated molecular and ozone transmittance, molecular backscatter coefficients and lidar instrument transmittance in elevation and geographic coordinates. And then compute the normalized signal;

Step 3: Evaluate the signal quality and atmospheric aerosol distribution, determine calibration range and horizontal average

distance, and screen the signals for calibration procedure;

Step 4: Calculate the calibration coefficients for the parallel and high-spectral-resolution channels, and then determine the calibration coefficients for the cross-polarized channel based on the polarization gain ratio.

Step 5: Obtain the global calibration coefficients by sliding averages in the along-track and neighboring-track directions using valid data;

Step 6: Compute the attenuated backscatter coefficient profiles.

The flow chart for the calibration is presented in Figure 2.

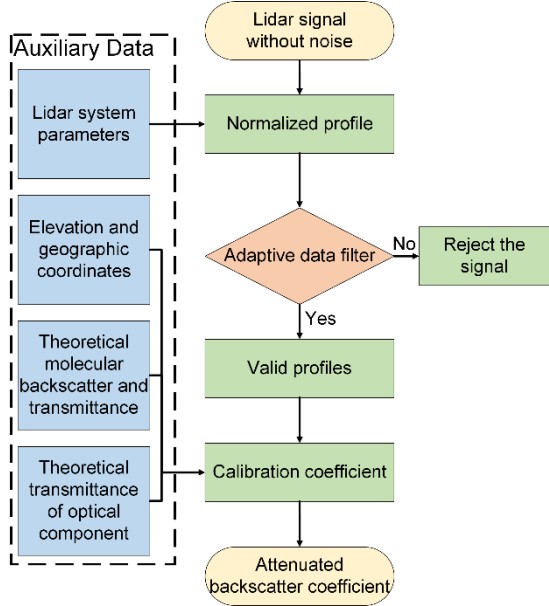

**Figure 2: Flow chart of ACDL calibration. The procedure for calculating molecular and ozone transmittance, molecular backscatter coefficients, and lidar instrument transmittance in elevation and geographic coordinates are illustrated in the blue boxes; Data**
**filtering is shaded orange; and the main calibration process are shaded green; Pre-calibration and calibrated data are shaded yellow.**



## 2.1 Theoretical basis and equations

The ACDL profiles are averaged vertically and horizontally on the satellite, with the averaging ratio depending on the altitude. After the geolocation and altitude corrections, the polarization and high-spectral-resolution channel data are obtained with a vertical resolution of 3 m at lower altitudes (below ~7.5 km), 24 m at higher altitudes (above ~7.5 km) along with a horizontal
resolution of about 0.33 km. The distance $r$ of the scatterer from the satellite can be expressed as

$$r = \frac{z_{sat}(k_p) - z(k_p)}{\cos(\theta(k_p))},$$
(1)

where $z$ is the height of the scatter above mean sea level, $z_{sat}$ is the satellite altitude, $k_p$ is the laser pulse index number, $\theta$ is the off-nadir angle.

Defining the normalized signals is the first necessary for the different channels including

$$X^{\mathrm{P}}(z, k_p) = \frac{r^2 P^{\mathrm{P}}(z, k_p)}{K^{\mathrm{P}} E_0(k_p) G_A} = C^{\mathrm{P}}(k_p) \beta^{\mathrm{P}}(z, k_p) T^2(z, k_p) f_{F-P}(z, k_p) , \text{ and}$$
(2)

$$X^{\mathrm{M}}(z, k_p) = \frac{r^2 P^{\mathrm{M}}(z, k_p)}{K^{\mathrm{M}} E_0(k_p) G_A} = C^{\mathrm{M}}(k_p) \beta^{\mathrm{M}}(z, k_p) T^2(z, k_p) f_{F-P}(z, k_p) f_I(z, k_p) ,$$
(3)

where $X$ is the signal normalized to laser energy and gain, $P$ is the received signal, $E_0$ is the laser pulse energy, $C$ is the calibration coefficient, $G_A$ is the amplifier gain, and $K$ is the system constant for each channel. The transmittance of F-P etalon and iodine vapor absorption filter are the function of height due to its dependence on atmospheric temperature and pressure,
which are denoted by $f_{F-P}$ and $f_I$. The superscript P represents the polarization channels, includes the parallel-polarized channel and the cross-polarized channel. And the superscript M represents the high-spectral-resolution channel. $T^2$ is the two-way transmittance of the laser in the atmosphere as a function of the length of the signal optical path, and therefore also as a function of the altitude (Bodhaine et al., 1999; Collis and Russell, 1976), and is given by

$$T^2(z, k_p) = \exp\left\{-2 \int_0^r \sigma[z(k_p), k_p] dr'\right\}.$$
(4)

Where $\sigma$ is the volumetric extinction coefficient, given by the following equation

$$\sigma(z, k_p) = \sigma_m(z, k_p) + \sigma_{O_3}(z, k_p) + \sigma_a(z, k_p) ,$$
(5)

with footnote $m, O_3, a$ are on behalf of molecular scattering, aerosol scattering, and ozone absorption.

For subsequent clarification, simplify the equations as

$$X^{\mathrm{P}} = C^{\mathrm{P}} \beta^{\mathrm{P}} T^2 f_{F-P} \text{ and}$$
(6)

$$X^{\mathrm{M}} = C^{\mathrm{M}} \beta^{\mathrm{M}} T^2 f_{F-P} f_I ,$$
(7)




Thus, the attenuated backscattering coefficient of the polarization and the high-spectral-resolution channel (hereafter referred to as multi-channel) could be derived by applying the calibration coefficients to the corresponding normalized signals, for different channels as

$$\beta'_\parallel(z,k_p) = \frac{X^\parallel(z,k_p)}{C^\parallel(k_p)} = \beta^\parallel(z,k_p)T^2(z,k_p) \,, \tag{8}$$

$$\beta'_\perp(z,k_p) = \frac{X^\perp(z,k_p)}{C^\parallel(k_p)PGR(k_p)} = \beta^\perp(z,k_p)T^2(z,k_p) \text{ and} \tag{9}$$

$$\beta'_M(z,k_p) = \frac{X^M(z,k_p)}{C^M(k_p)} = \beta^M(z,k_p)T^2(z,k_p) \,. \tag{10}$$

Where the $\parallel$ represents the parallel channel, $\perp$ represents the cross-polarized channel. And the PGR (polarization gain ratio) is a conversion factor that quantifies the relative magnitudes of the parallel- and cross-channel detector gains, detector quantum efficiencies, amplifier gains, and optical efficiencies downstream of the polarization beam splitter (Hunt et al., 2009; Alvarez et al., 2006).

**2.2 Calibration procedure**

The ACDL 532 nm multi-channel calibration coefficients for nighttime conditions using the molecular normalized technique. The technique requires that the backscatter in the calibration region comes primarily from molecules. To estimate the calibration coefficients, the calculated ratio is based on the normalized signal and the modelled attenuated backscatter (Russell et al., 1979; Hostetler et al., 2006; Reagan et al., 2002). The ACDL selects 31–35 km as calibration region. The following subsections provide a detailed explanation of the mathematical basis for the calibration procedure.

Since there are still a small amount of aerosols present in the calibration regions (Vernier et al., 2009), the relative contribution of aerosol backscattering is evaluated by utilizing the aerosol data from Stratospheric Aerosol and Gas Experiment-III (SAGE III, Cisewski et al., 2014). The 532 nm parallel-channel and high-spectral-resolution channel calibration coefficient equations are formed to solve for $C^\parallel$ and $C^M$ as

$$C^\parallel = \frac{X^\parallel(z_c)}{\hat\beta^\parallel(z_c)\hat T^2(z_c)\hat R^\parallel \hat f_{F-P}(z_c)} \text{ .and} \tag{11}$$

$$C^M = \frac{X^M(z_c)}{\hat\beta^M(z_c)\hat T^2(z_c)\hat f_{F-P}(z_c)\hat f_I(z_c)} \,. \tag{12}$$

Where $X^\parallel$ and $X^M$ are the normalized signal measured by ACDL, the ^ superscript denotes the parameters estimated from atmospheric model, and $z_c$ is the designated altitude. Global temperature, ozone mass mixing ratio and pressure data are obtained from the ERA5 dataset (Hersbach et al., 2020). The ERA5 dataset contains 37 barometric pressure levels and provides hourly averaged global atmospheric parameters on a 0.25° latitude × 0.25° longitude grid. The ERA5 global data is aligned with the altitude, latitude and longitude of the ACDL profiles. which,



$$\hat{R}^{\parallel} = \frac{\hat{\beta}^{\parallel}(z_c)}{\hat{\beta}_m^{\parallel}(z_c)} = \frac{\hat{\beta}_m^{\parallel}(z_c) + \hat{\beta}_a^{\parallel}(z_c)}{\hat{\beta}_m^{\parallel}(z_c)}. \tag{13}$$

In Eq. (13), the total backscattering coefficient of the parallel channel is subdivided into molecular volume scattering and aerosol volume scattering, with $\hat{\beta}_m^{\parallel}(z_c)$ and $\hat{\beta}_a^{\parallel}(z_c)$ are the parallel component of the molecular and the aerosol volume backscatter coefficient, respectively. The parallel component of molecular backscatter is calculated from estimations of the total molecular backscatter $\hat{\beta}_m$ and the expected depolarization ratio for molecular backscatter $\delta_m$.

$$\hat{\beta}^{\parallel}(z_c) = \frac{1}{1+\delta_m}\hat{\beta}_m(z_c) = 0.996\hat{\beta}_m(z_c), \tag{14}$$

with,

$$\delta_m = \frac{\hat{\beta}_m^{\perp}}{\hat{\beta}_m^{\parallel}} = 0.00366. \tag{15}$$

The bandwidth of the F-P narrowband filter used in the ACDL is less than 30 pm, so 0.00366 was chosen as the ratio of cross to parallel backscatter as the central Cabannes line where the backscatter can be detected (She, 2001; Cairo et al., 1999).

The total molecular backscatter is obtained by calculating the product of the molecular number density and the total Rayleigh scattering cross section for air mass (Reagan et al., 2002; Cairo et al., 1999), by the following Eq. (16)

$$\hat{\beta}_m = \frac{\hat{\sigma}_m(z_c)}{S_m} = \frac{\hat{\sigma}_m(z_c)}{(\frac{8\pi}{3})k_{bw}}, \tag{16}$$

ACDL has selected a widely used value of 8π/3 for the lidar ratio to calculate molecular, as $S_m = (\frac{8\pi}{3})k_{bw}$. This value is commonly used in the lidar community (Collins and Russell, 1976). And $k_{bw}$=1.0401 defines the dispersion of the refractive index and the King correction factor of air at 532 nm (She 2001; Hostetler et al., 2006; Reagan et al., 2002).

The molecular volume scattering coefficient $\hat{\sigma}_m$ can be calculated at the corresponding altitude $z_c$ by

$$\hat{\sigma}_m(z_c) = \frac{N_A P(z_c) Q_S}{R_a T(z_c)}, \tag{17}$$

At the altitude $z_c$, ERA5 provides the pressure $P(z_c)$ (in hPa) and the temperature $T(z_c)$ (in K) (Hersbach et al., 2020). The Avogadro's number $N_A$ is 6.02214×10²³ mol⁻¹, the gas constant $R_a$ is 8.314472 J·K⁻¹mol⁻¹, and the cross section for total Rayleigh scattering per molecule for 532 nm $Q_S$ is adopted as 5.167×10⁻²⁷ cm² (Hostetler et al., 2006; Bucholtz, 1995).

The two-way signal attenuation $\hat{T}^2$ is defined as the attenuation of the signal from lidar transmitter to the scattering volume and back to the receiver. Different from Eq. (4), the attenuation for the ACDL to the calibration altitude $z_c$ can be described with

$$\hat{T}^2(z_c) = \exp\left\{-2\int_{z_{sat}}^{z_c}[\hat{\sigma}_m(z') + \hat{\sigma}_{O_3}(z') + \hat{\sigma}_a(z')]dz'\right\}, \tag{18}$$

Where $\hat{\sigma}$ with the footnote $m, O_3, a$ represents the extinction coefficients of molecular, aerosol and ozone The $\hat{\sigma}_{O_3}$ is given by



$$\hat{\sigma}_{O_3} = c_{O_3}\hat{\varepsilon}_{O_3}, \tag{19}$$

where $c_{O_3}$ is the Chappius ozone absorption coefficient in m$^{-1}$. The ozone absorption coefficient is obtained at the correct wavelength from empirical table (Yorks et al., 2016; Iqbal, 1984; Vigroux, 1953). The $\varepsilon_{O_3}$ is the column density for ozone mass mixing ratio conversion, calculated by the following equation of

$$\hat{\varepsilon}_{O_3}(z_c) = \frac{r_{O_3}(z_c)P(z_c)}{2.14148\times10^{-5}RT(z_c)}. \tag{20}$$

The ozone mass mixing ratios $r_{O_3}$ are firstly converted to column density per kilometer (atmcm/km, Hersbach et al., 2020),

the gas constant $R$ is 287.058 J·K$^{-1}$kg$^{-1}$. The transmittance curves calculated from the above Eq. (18) are shown in Figure 3.

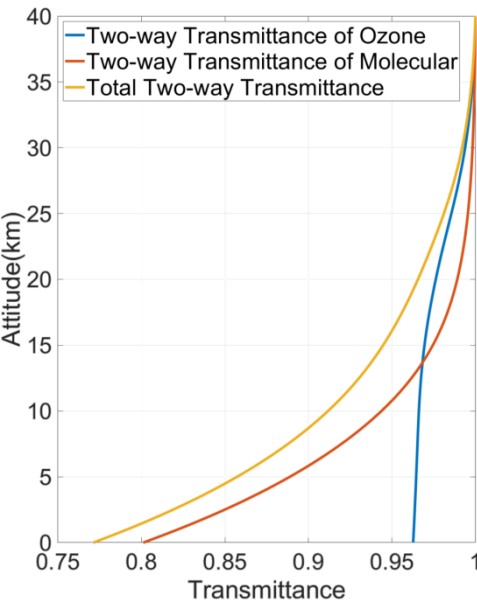

**Figure 3: Two-way transmittance of ozone (blue line), molecules (orange line) and total two-way transmittance (yellow line).**

Due to the variation of atmospheric molecular broadening at the calibration altitude with temperature and pressure, the transmittance of signals within both the F-P etalon and the iodine vapor absorption filter also fluctuates. The transmittance is

calculated under different temperature and pressure conditions. The following are the Rayleigh scattering functions of the molecular signals through the F-P etalon and the iodine vapor absorption filter (Flesia and Korb, 2000):

$$\hat{f}_{F-P}(T,P,\nu') = \int \mathcal{R}_m(T,P,\nu')\hat{F}_{F-P}(\nu')d\nu' \quad \text{and} \tag{21}$$

$$\hat{f}_I(T,P,\nu') = \int \mathcal{R}_m(T,P,\nu')\hat{F}_I(\nu')d\nu'. \tag{22}$$

Where $\mathcal{R}_m$ is the normalized Rayleigh scattering function, $\nu$ is the frequency of the backscattering signal of the molecules at

the calibration heights, and $\hat{F}_{F-P}$, $\hat{F}_I$ are the transmittance functions of the F-P etalon and the iodine molecular absorption filter calibrated in the laboratory. The iodine molecular absorption filter of ACDL use iodine absorption line 1110 (Dong et





al., 2018), the measured transmittance spectrum as shown in the following Figure 4. As demonstrated in Figure 4, the molecular broadening at heights of 30-40 km extends across the absorption line 1109. Consequently, the effect of the 1109 line was rectified in the transmittance calculation. The F-P transmittance curve completely covers the scattering spectra of the molecules in the calibration region, so there is no need for an additional schema in Figure 4.

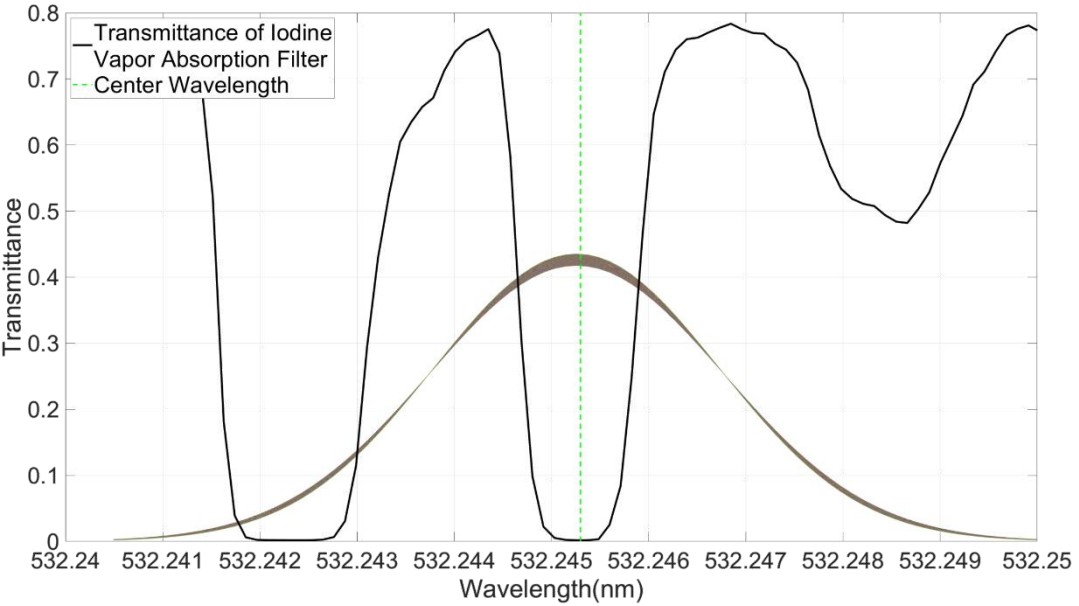

**Figure 4: Transmittance function (black line) of the iodine vapor absorption filter (blue dotted line means the center wavelength, and multiple color curves means normalized Rayleigh scattering function at 30–40 km).**

The calibration for the cross-polarized channel requires the application of the polarization gain ratio $PGR$ of the cross-polarized channel to the parallel channel, which defined as,

$$PGR \equiv \frac{c^\perp}{c^\parallel} \approx \frac{G^\perp}{G^\parallel},\tag{23}$$

$G^\parallel$ and $G^\perp$ are overall responsivity and gain of the parallel channel and the cross channel (Hostetler et al., 2006).

Before commencing the calibration process, it is essential to determine the heights of the calibration region. The selection of the calibration heights for ACDL is guided by the following principles: firstly, the signal-to-noise ratio of the signal in the upper atmosphere is low, necessitating substantial data averaging; secondly, the lower atmosphere is significantly impacted by aerosols, which are unsuitable for calibration heights that should only comprise of molecular scattering (Kar et al., 2019; Kyrölä et al., 2013; Wang et al., 2020). Taking the ACDL data frame range and signal quality into account, the calibration region was set between 31 and 35 km. This altitude range in the stratosphere is sufficiently high to be relatively free of aerosols (albeit not completely so), and low enough to ensure a backscatter signal of adequate magnitude, given the mean molecular number density.



The ACDL averages the high-altitude data on the satellite and downloads the profile data with a horizontal resolution of 330 m and a vertical resolution of 24 m. However, the data quality in this instance fails to satisfy the calibration requirements, so a large amount of additional data averaging is required to obtain an accurate calibration coefficient estimate. Firstly, the data from the calibration region are averaged horizontally at 3.6 km intervals. Then, these averaged data profiles are transformed

into provisional calibration coefficient composite profiles by using Eqs. (28) and (29). Finally, the calibration coefficients are further averaged through a 150-point sliding average, providing the effective 500-km average between independent samples. In summary, the equation for calculating the calibration coefficients for the nighttime 532 nm parallel and high-spectral-resolution channel are as follow:

$$\tilde{C}(y_k) = \frac{1}{j_{31km} - j_{35km} + 1} \sum_{j=j_{31km}}^{j=j_{35km}} \frac{\frac{1}{11} \sum_{11k-5}^{11k+5} X(z_j, y_i)}{\hat{\beta}(z_j, y_k) \hat{R}(z_j, y_k) \hat{T}^2(z_j, y_k) \hat{f}_{F-P}(z_j, y_k)}, \tag{24}$$

$\tilde{C}(y_k) = \frac{1}{139} \sum_{k-69}^{k+69} \tilde{C}(y_k)$ and     (25)

Where $i$ and $j$ are the index for horizontal and vertical sample in one profile, $y$ and $z$ are the horizontal distance and vertical distance along the track. The ~ superscript denotes the parameters that are smoothed every 500 km along the track.

Figure 5b and 5c illustrate the calibration coefficient $C$ as a function of the smoothed calibration coefficient $\tilde{C}$ along latitude for the parallel channel and the high-spectral-resolution channel, respectively. An example with the orbit of 9928 on July 1st,

2022 is presented in Figure 5.

The correlation between the lidar received signal and the atmospheric model of ERA5 at the range of 31 and 35 km increases significantly as the mean distance increases. However, increasing the mean distance is also accompanied by a significant increase in the variation of molecular backscattering coefficients computed by the atmospheric model, particularly at high latitudes. The stratospheric aerosol content between 31 and 35 km is non-uniform, resulting in inaccurate characterization of

calibration coefficients in areas affected by aerosols and introducing additional errors. And the impact of high-energy events in certain areas (e.g. South Atlantic Anomaly, SAA, Hunt et al., 2009) can spread over a large region due to the long averaging distances. To accurately calculate the global ACDL calibration coefficient, additional sliding average of 500 km in the direction of adjacent-track distances. This method requires a specific duration (usually one month) to accumulate data. During the initial data processing phase, only the 500 km along-track average is used to compute the daily calibration coefficients. Later, the

globally averaged calibration coefficient is applied to recalibrate both the processed and unprocessed ACDL raw data during the accumulation criteria for recalibration.



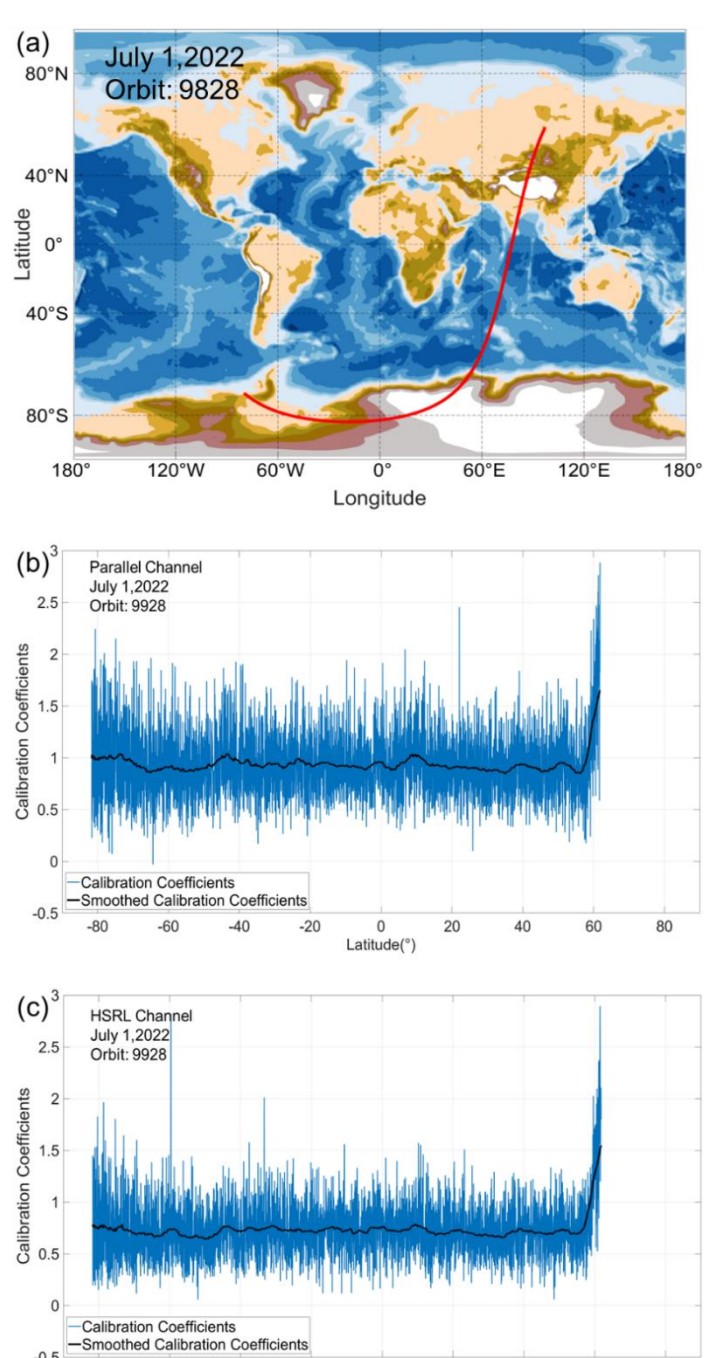

**Figure 5: The calibration coefficients for each calibration region. (a) The example orbit 9928 for July 1ˢᵗ, 2022; (b) The estimated parallel channel calibration coefficients from the 3.6 km average (blue line) and the smoothed calibration coefficient results after 500 km sliding average (black line) are displayed. (c) The estimated high-spectral-resolution channel calibration coefficients from the 3.6 km average (blue lines) and the smoothed calibration coefficient results after 500 km sliding average (black line) are presented. The orbital track segment that corresponds to calibration coefficients can be found in the upper right corner of the figure.**




**2.3 Adaptive data filter**

The lidar data contains random signal spikes, which can significantly impact the calibration coefficients (Lee et al., 2008).
These high-energy events are mainly concentrated in the SAA region, but also occur randomly throughout the detection (Hunt et al., 2009). Thus, it is necessary to filter the collected data to exclude spikes from the data used for calibration.

The calibration procedure for ACDL involves a filtering technique that consists of three sequential steps to filter the signal extracted in the previous step.

Step 1: Each signal in the calibration area (altitude range of 31-35 km and horizontal range of 3.6 km) is first filtered using
upper and lower limits based on the theoretical values X, as well as the fluctuation of the measured profiles. The Eqs. (26) are used to determine the thresholds of $X_T$ for the parallel channel and the high-spectral-resolution channel, respectively.

$$X_T = \hat{X}(l) \pm k_m \Delta X , \tag{26}$$

Where $\hat{X}(l)$ is the theoretical value of $X(z_k)$ at position $l$, which corresponding to the latitude of the signal profile. The subscript $T$ denotes the threshold range of the filter. The empirical definition of scaling factors $k_m$ varies across channels and
can be adjusted. Theoretical profiles are estimated from the modelled molecular number density within the calibration region. The random uncertainties $\Delta X$ consider the random errors (shot noises) in the signals. The equation below defines the $\Delta X$ as a function of latitude,

$$\Delta X = \sqrt{\frac{\sum\left((X(z_j,y_k) - \hat{X}(z_j,y_k)) - \frac{X(z_j,y_k) - \hat{X}(z_j,y_k)}{n}\right)^2}{n}}. \tag{27}$$

Where $z_j$, $y_k$ is the location of the calibration profile, $n$ is the number of the averaged bins. $\Delta X$ represent the standard deviation
(Std) of the differences between the actual measured values and the theoretical values.

Step 2: The second step of the filtering technique involves the use of the Noise-to-Signal Ratio (NSR, Lee et al., 2008; Powell et al., 2009) test to evaluate signals in the pre-calibration area between 31–35 km vertically and 3.6 km horizontally. This test determines whether there are significant variations in signal magnitude within the specified region. To calculate NSR, use the formula below:

$$NSR = \frac{\sigma_{std}(X_{valid})}{\mu(X_{valid})}. \tag{28}$$

Where $\sigma_{std}$ is the Std and $\mu$ is the mean value of the valid signals $X$. The resulting values are compared to an empirically defined NSR. Any high-energy events exceeding the NSR threshold get excluded from the calibration procedure. Signals that are valid within the calibration range that has been accepted are then entered into the next step of the calibration process. If the entire profile is excluded, the neighboring coefficients are utilized to determine the daily estimated calibration coefficient at
the position.





**Figure 6: Schematic representation of the signal screening by NSR. (a) An example orbit passing through the SAA region, orbit 9808; (b) The parallel channel NSR as a function of the corresponding latitude, orbit 9808. The dotted line indicates the NSR threshold**





**value of 3.23. The 11 $X^{\parallel}$ signal profiles located near the (d) edge and (f) centre of SAA; (c) The high-spectral-resolution channel NSR**
**as a function of the corresponding latitude, orbit 9808. The dotted line indicates the NSR threshold value of 3.15. The 11 $X^{M}$ signal**
**profiles located near the (e) edge and (g) centre of SAA. The valid signals detected by the filter, as indicated by the shaded areas.**

The NSR values along the latitude in the parallel and high-spectral-resolution channels are illustrated in Figure 6 for the same

orbit (as seen in Figure 6a). The NSR threshold is set to 3.23 for the parallel channel and to 3.15 for the high-spectral-resolution

channel, respectively. Additionally, Figure 6b and 6c demonstrate the application of the NSR test within calibration regions.

Figure 6d to 6g display the comparies of signals and high-energy events between the centre and edge profiles of the SAA. The

adaptive filter identifies valid signals within the shaded regions, and exclude the data that falls outside of the shaded range.

The remaining profiles were utilized as valid data in the subsequent stage of the averaging calculation.

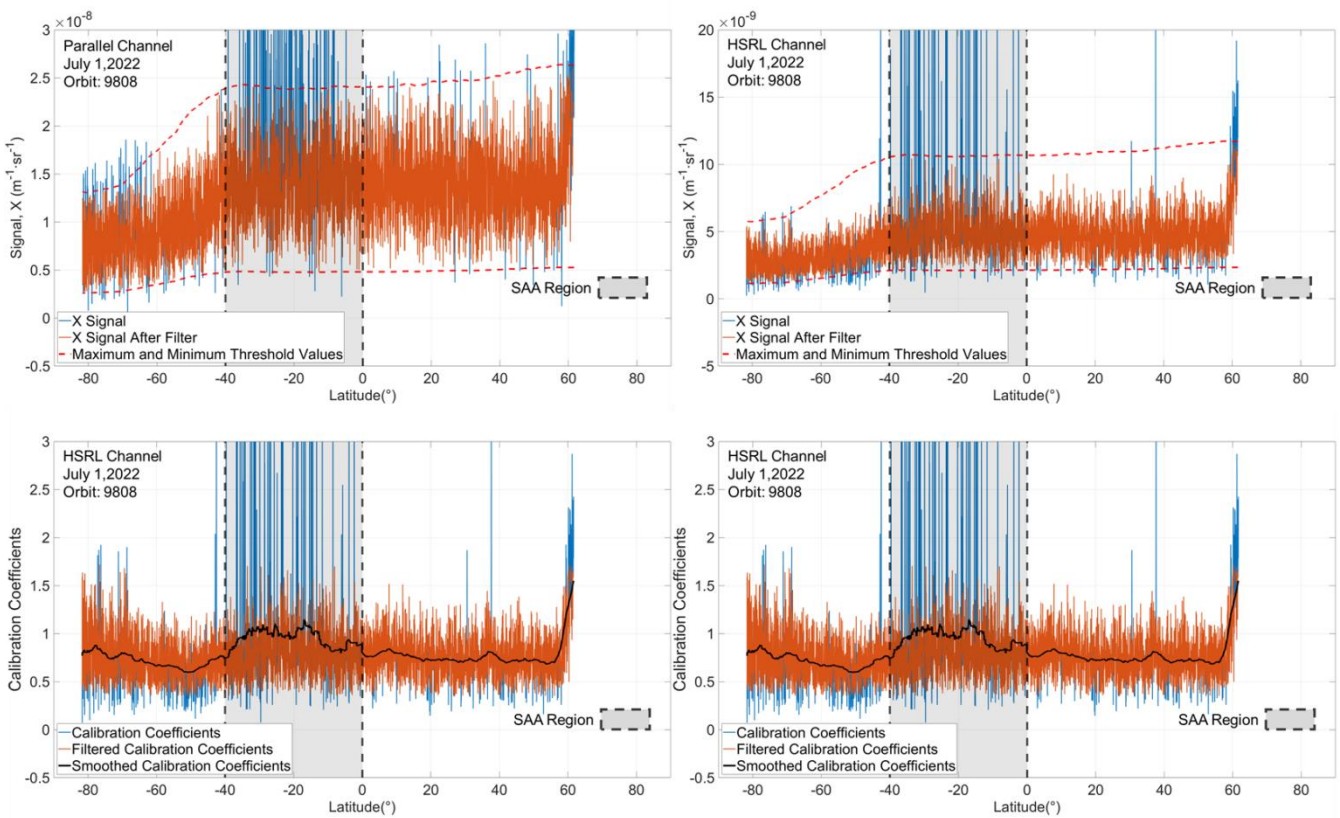

**Figure 7: Schematic of the original signal and calibration coefficients after filtering (orbit 9808). (a) The average signal $X^{\parallel}$ as a**
**function of the corresponding latitude for altitudes between 31 and 35 km, 1 July 2022. Within the SAA, there is a significant**
**variation in the original signal, as indicated by the blue lines. The adaptive filter defines the minimum and maximum values with**
**dotted lines, and the orange lines show the signals after filter. And the red dotted lines indicate the range of thresholds; (b) The**
**average signal $X^{M}$ as a function of the corresponding latitude for altitudes between 31 and 35 km, orbit 9808. The lines in Figure 7b**
**have the same meaning as in Figure 7a; (c) The filtered (orange) and unfiltered (blue lines) calibration coefficients of the parallel**
**channel. The black line plots the smoothed calibration coefficients; (d) The filtered (orange) and unfiltered (blue lines) calibration**
**coefficients of the high-spectral-resolution channel. The black line plots the smoothed calibration coefficients.**

Step 3: In the third step, the mean of the profiles that can be calibrated is filtered using threshold determined by Eqs. (32) and

(33) above. The final filter evaluates whether the mean values of the signal profiles available for the calibration calculation





fall within the pre-set threshold range. If some profiles are rejected, the nearest calibration coefficients are selected as the daily
calibration coefficient estimates for this area.

Figure 7a and 7b examplify the application of the adaptive filter, depicting the $X^{\parallel}$ and $X^M$ signals as a function of latitude before and after the screening. The data pertains to the 9808 orbit 31–35 km altitude average on July $1^{st}$, 2022. As depicted in the graph, the signal demonstrates significant fluctuations while passing through the SAA region, which lies spans from the equator to about 40°S. High-energy events at the calibration regions create remarkably high signal spikes, but the ACDL
measured signal results in less pronounced negative signal spikes. Figure 7c and 7d depict the screened profiles, displaying the calibration coefficients before and after the data filtering. The results indicate that the adaptive filter effectively removes the errors caused by high-energy events when calculating calibration coefficient, allowing the ACDL calibration procedure to accurately determine 532 nm calibration coefficients.

## 3. Assessment

Assessments are continuously conducted on the ACDL nighttime 532 nm multi-channel calibration procedure during the mission. The assessment is evaluated through error analysis and validation. The error analysis assesses systematic and random errors, and validation is achieved by utilizing attenuated backscatter coefficients and pure-molecule attenuated scattering ratios. Validation tests, such as comparing airborne lidar observations and ground-based lidar networks, will be carried out in the further research.

### 3.1 Error analysis

The uncertainty of the calibration coefficients comprises systematic and random errors, which can be expressed as,

$$\left(\frac{\Delta C}{C}\right)^2 = \left(\frac{\Delta C}{C}\right)^2_{System} + \left(\frac{\Delta C}{C}\right)^2_{Random}. \tag{29}$$

The systematic uncertainty component of the parallel channel is given by

$$\left(\frac{\Delta C^{\parallel}}{C^{\parallel}}\right)^2_{System} \approx \left[\frac{\Delta \hat{R}^{\parallel}(z_c)}{\hat{R}^{\parallel}(z_c)}\right]^2 + \left[\frac{\Delta \hat{\beta}^{\parallel}_m(z_c)}{\hat{\beta}^{\parallel}_m(z_c)}\right]^2 + \left[\frac{\Delta \hat{T}^{\parallel}(z_c)}{\hat{T}^{\parallel}(z_c)}\right]^2 + \left[\frac{\Delta \hat{T}_{F-P}(z_c)}{\hat{T}_{F-P}(z_c)}\right]^2 + \left[\frac{\Delta \hat{E}(z_c)}{\hat{E}(z_c)}\right]^2. \tag{30}$$

Table 2 presents the estimates of the systematic error components for the ACDL nighttime 532 nm calibration procedure. The source of uncertainty analysis should reasonably apply to calibration regions between 31 and 35 km in the stratosphere. As more accurate information is acquired on the precision of the products used for calculations, estimations of the error terms contributing will be enhanced. Currently, these diverse components create a comprehensive relative systematic error of ~5% for $\Delta C^{\parallel}/C^{\parallel}$. The error resulting from the two-way transmittance $T^2$ is negligible, at less than 0.005%, and is disregarded in the
calculations conducted.

**Table 2: Systematic error components for the 532 nm parallel channel calibration coefficient.**

| $\left(\frac{\Delta C^{\parallel}}{C^{\parallel}}\right)_S$ | $\frac{\Delta R^{\parallel}(z_c)}{R^{\parallel}}$ | $\frac{\Delta \hat{\beta}^{\parallel}_m(z_c)}{\hat{\beta}^{\parallel}_m(z_c)}$ | $\frac{\Delta \hat{T}_{F-P}(z_c)}{\hat{T}_{F-P}(z_c)}$ | $\frac{\Delta \hat{E}(z_c)}{\hat{E}(z_c)}$ |
| --- | --- | --- | --- | --- |





| 0.045 | 0.03 (Cisewski et al., 2014) | 0.03 (Hersbach et al., 2020) | 0.01 | 0.01 |

The systematic uncertainty component of high-spectral-resolution channel is given by

$$\left(\frac{\Delta C^M}{C^M}\right)^2_{System} \approx \left[\frac{\Delta\hat{\beta}_m^M(z_c)}{\hat{\beta}_m^M(z_c)}\right]^2 + \left[\frac{\Delta\hat{T}^M(z_c)}{\hat{T}^M(z_c)}\right]^2 + \left[\frac{\Delta\hat{T}_{F-P}(z_c)}{\hat{T}_{F-P}(z_c)}\right]^2 + \left[\frac{\Delta\hat{T}_I(z_c)}{\hat{T}_I(z_c)}\right]^2 + \left[\frac{\Delta\hat{E}(z_c)}{\hat{E}(z_c)}\right]^2, \tag{31}$$

The error evaluation of the high-spectral-resolution channel follows the same method, with Table 3 listing the current best
estimations of the systematic error components. Presently, the equation above can be used to calculate the overall relative systematic error $\Delta C^M/C^M$ is ~4%.

**Table 3: Systematic error components for the 532 nm high-spectral-resolution channel calibration coefficient.**

| $\left(\frac{\Delta C^M}{C^M}\right)_S$ | $\frac{\Delta\hat{\beta}_m^M(z_c)}{\hat{\beta}_m^M(z_c)}$ | $\frac{\Delta\hat{T}_{F-P}(z_c)}{\hat{T}_{F-P}(z_c)}$ | $\frac{\Delta\hat{T}_I(z_c)}{\hat{T}_I(z_c)}$ | $\frac{\Delta\hat{E}(z_c)}{\hat{E}(z_c)}$ |
|---|---|---|---|---|
| 0.035 | 0.03 (Hersbach et al., 2020) | 0.01 | 0.01 | 0.01 |

The averaged random uncertainty $\Delta C(y_k)$ is given by

$$\Delta C = \sqrt{\frac{\sum\left(C(y_k) - \hat{C}(z_j, y_k)\right)^2}{n}}. \tag{32}$$

The random uncertainty is estimated by calculating the Std of the calibration coefficients. In the estimation process of the random uncertainty, only the error attributable to the calculated calibration coefficients itself is considered. More influences of the error attributable to the calibration coefficients are further considered in the subsequent evaluation process.

The uncertainty in calibration coefficient for the cross-polarized channel can be calculated by considering the error in parallel channel and the error in polarization gain ratio, as shown in the following equation (Powell et al., 2009):

$$\left(\frac{\Delta C^\perp}{C^\perp}\right)^2 = \left(\frac{\Delta C^\parallel}{C^\parallel}\right)^2 + \left(\frac{PGR}{PGR}\right)^2. \tag{33}$$

The $PGR$ is currently determined to have a measurement error of ~1%, and the estimation includes both random and systematic errors.

## 3.2 Validation

The molecular normalization technique relies on matching the signal in the purely molecular atmospheric region at high altitude,
to achieve the calibrations for all the individual profiles. To accurately assess the calibration coefficients, the matching of the attenuated backscatter coefficients in the calibration region with the model must be determined first. Figure 8 presents the results of 31–35 km calibrated averaged signals.





**Figure 8: Average for the measured signal (blue line), model estimates (orange line), and calibrated signal (yellow line) along the latitude. (a) and (b): These values for the parallel channel on July 30th and October 31st; (c) and (d): These values for the high-spectral-resolution channel on July 30th and October 31st; (e) and (f): the orbits of DQ-1 nighttime measurement for July 30th and October 31st, 2022. The estimated average values were calculated over a vertical distance of 31–35 km and a horizontal sliding average of 100 km.**

After the calibration procedure, the signal *X* was corrected to align with the modeled attenuation backscatter coefficient, as demonstrated in Figure 8 on July 30th and October 31st, 2022. The results indicate that the calibrated backscatter coefficients have a total relative error of less than 2% comparing with the mean value of the modeled results, achieve a satisfactory match within the range of 31–35 km. The relative error is calculated using the following formula:



$$\delta X(z_{31-35},\ l) = \frac{X(z_{31-35},l) - \hat{X}(z_{31-35},l)}{X(z_{31-35},l)} \times 100\%, \tag{34}$$

where $X$ is the mean of calibrated attenuated backscatter coefficient at latitude $l$ at the range $z$ of 31–35 km, $\hat{X}$ is the theoretically attenuated backscatter coefficient from model, and $\delta X$ is the relative error.

The performance of calibration procedure can also assess by calculating the pure-molecule attenuated scattering ratios using attenuated backscatter data of ACDL and comparing them to the theoretical pure-molecule scattering ratio of 1. Regions with extremely low aerosol loading are referred as pure molecular region. Previous studies have demonstrated that the clear-air area between 8 and 12 km in altitude typically loaded limited aerosol, which leading to a total scattering ratio observed to be close to 1 (Vaughan et al., 2004 & 2009). However, 8-12 km is quite a low altitude, and it is difficult to ensure that the assumption that the 12-40 km range contains no aerosols is valid even under clear-air conditions. Therefore, ACDL adopts 26–30 km as a pure molecular region to calculate the attenuated scattering ratio to validate the multi-channel calibration algorithm. At low aerosol contents, the difference between the pure-molecule scattering ratios calculated from the calibrated attenuated backscatter coefficients and the molecular backscatter estimate should be less than ~5%, which is the relative calibration uncertainty. Through this comparison, it is possible to identify the existing bias in the ACDL calibration.

The attenuated pure-molecule scattering ratios $R'$ is defined as

$$R'_{PM}(z_{26-30}, k) = \frac{\beta'(z_{26-30}, k)}{\hat{\beta}_m(z_{26-30}, k)}. \tag{35}$$

where $k$ is the index of profile and $z_{26-30}$ denotes the altitude range used to calculate the scattering ratio. The measured attenuated backscatter coefficients are determined by molecular backscatter, faint aerosol backscatter, and extinction within the atmosphere, while the modeled attenuated scattering ignores the effect of aerosol (Vaughan et al., 2004; McGill et al., 2007). Expanding Eq. (35) yields

$$
\begin{aligned}
R'_{PM}(z_{26-30}, k) &= \frac{[\beta_m(z_{26-30},k)+\beta_a(z_{26-30},k)]T_m^2(z_{26-30},k)T_{O_3}^2(z_{26-30},k)T_a^2(z_{26-30},k)}{\hat{\beta}_m(z_{26-30},k)\hat{T}_m^2(z_{26-30},k)\hat{T}_{O_3}^2(z_{26-30},k)} \\
&= \left[\frac{\beta_m(z_{26-30},k)+\beta_a(z_{26-30},k)}{\hat{\beta}_m(z_{26-30},k)}\right]T_a^2(z_{26-30},k) \\
&= \left[1 + \frac{\beta_a(z_{26-30},k)}{\hat{\beta}_m(z_{26-30},k)}\right]T_a^2(z_{26-30},k)
\end{aligned}
. \tag{36}
$$

After completing the signal calibration, the difference between the calculated pure-molecule scattering ratio and the estimated molecular backscatter is within 5% uncertainty at the 26–30 km region, assuming negligible aerosol attenuation between 30–40 km under pure-molecule conditions. To reduce the effect of noise, the profile of the pure molecular attenuated scattering ratio $\langle R'_{CA}(z_{26-30}, k)\rangle$ was averaged over ~200 km (Powell et al., 2009) segment by using Eq. (37) (as illustrated in Figure 9)

$$R'_{PM200} = \frac{1}{600}\sum_{k=1}^{600}\langle R'_{PM}(z_{26-30}, k)\rangle. \tag{37}$$



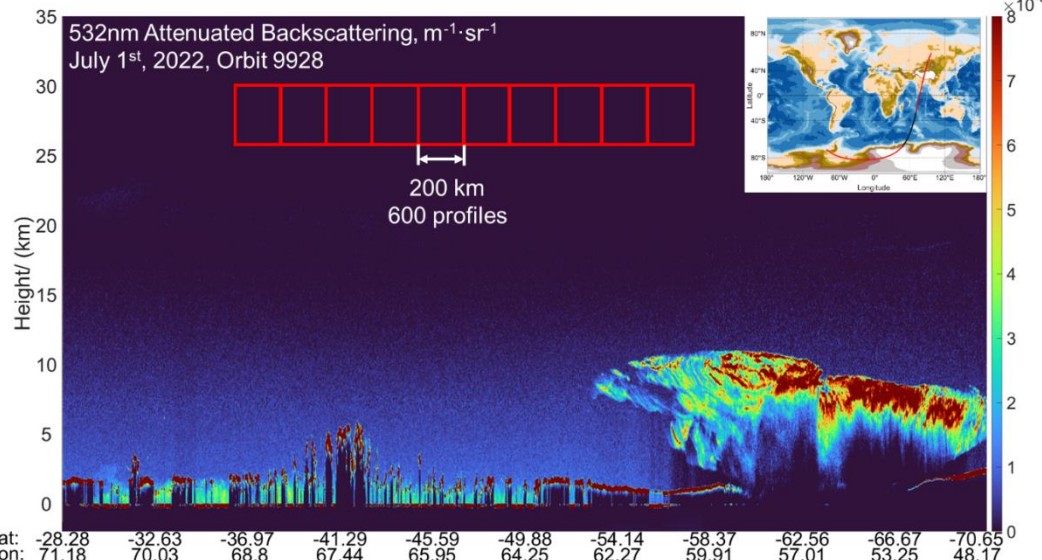

**Figure 9: Lidar 532 nm total attenuated backscatter coefficient (m⁻¹sr⁻¹, 1 July 2022). The clear-air regions are illustrated by red boxes, spanning 200 km in length and ranging from 26 to 30 km in altitude. The upper right figure displays the range of track (black line).**

Figure 10 illustrates four consecutive profiles of the pure-molecule attenuated scattering ratio $R'_{PM}(z_{26-30}, k)$ for polarization and high-spectral-resolution channels from July 1$^{st}$, 2022. The profiles display the ratio fluctuations and are at single profile resolution. A composite profile was produced by averaging twelve profiles (330-m horizontal resolution) over the altitude range of 26 to 30 km. And the horizontal distance between profiles is approximately 4 km. The dashed line is present in each plot to represents the scattering ratio of 1.0 for reference. The average attenuated scattering ratio for profiles is represented by the vertical solid lines. In these cases, the average attenuated scattering ratios deviate partially from the expected 1±0.05 due to the shot noise of the single and the impact of the faint aerosol.





**Figure 10: pure-molecule attenuated scattering ratio profiles (blue lines) from July 1st, 2022. The expected scattering ratios of 1 (black dotted lines), and the mean value of the ratios (black solid lines). (a) Profiles for the polarization channel; (b) Profiles for the high-spectral-resolution channel.**

Figure 11 displays the mean of 600-profiles (~200 km, Powell et al., 2009) for pure-molecule attenuated scattering ratio $\langle R'_{PM}(z_{26-30}, k) \rangle$ at a 330-m horizontal resolution. And averaging the mean of 600-profiles, $R'_{PM200}$ for polarization channel is about 1, and for high-spectral-resolution channel is about 0.98. Although the mean ratio for single profile was not as expected, the average over 200 km kept the mean attenuation scattering ratio results within the range of 1±0.05. The Std as shown in Figure 11 depicting that the fluctuations primarily stem from the shot noise.



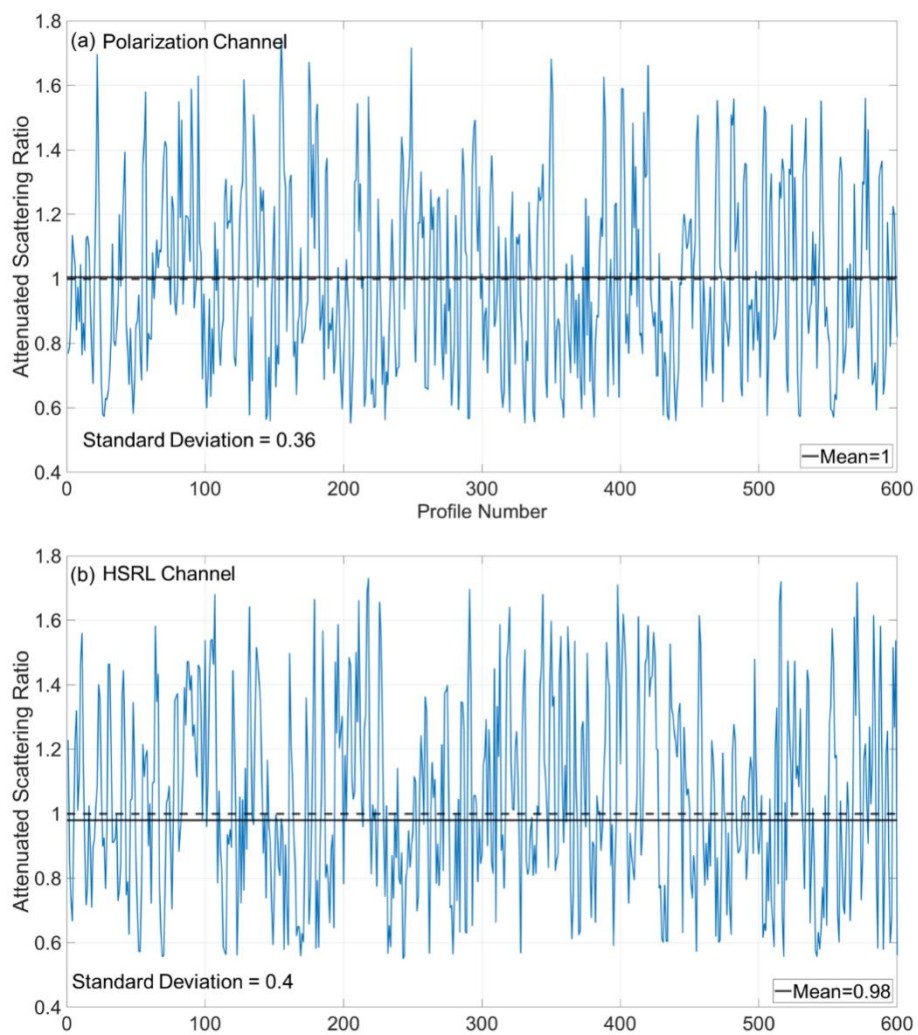

**Figure 11: The mean of the averaged pure-molecule attenuated scattering ratio for 600-profiles (330-m horizontal resolution, ~200 km segment, blue line). The mean of averaged pure-molecule scattering ratio (black solid line), and the expected value of 1.0 (black dotted line). (a) Ratios for the polarization channel; (b) Ratios for the high-spectral-resolution channel.**

## 4. Discussion and outlook

The molecular normalized calibration technique applied to ACDL has been successfully applied to its polarization channel, and demonstrates its feasibility for high-spectral-resolution channel. Planned algorithm improvements include updating the adaptive filter and further removing the effects of dark noise.

Figure 12 illustrated the result of global calibration coefficient on a 1° latitude × 1° longitude grid for July, 2022. The Arctic and adjacent regions are in the polar day range in July, so there are no calibration coefficients for nighttime. During nighttime measurements, the strong backscattering targets (such as Tibetan Plateau and Antarctica) produced higher intensity of signals




than expected. Also visible are regions with increased calibration coefficients near the pole, which are caused by an increase

in dark noise (Hunt et al., 2009). The ACDL global dark noise is unstable due to the absence of reliable dark noise detectors. This issue is further intensified by the presence of shot noise when a strong backscatter target is detected, increased challenging in data processing. Despite the current calibration procedure use the denoised lidar signal, and the filter has reduced the effects of these events, the fluctuations in dark noise still cause deviations in the calibration coefficients. The influence of these events also spreads as the sliding average frame progresses. Evaluate the impacts of different dark noise feature for lidar signals at

night and rectify calibration coefficients in such regions will be conducted in the follow-up study.

In the future, the ACDL scientific team plans to continuously conduct validation tests to verify the reliability of the calibration algorithm. This will involve adding simultaneous observations using various types of lidar and extending the range of validation. In the future development of the enhanced instrument, the scientific team will fully consider the issues of dark noise and strong backscattered signals. The instrument proposed improvements include increasing the reception of dark noise signals

and reducing background noise level.

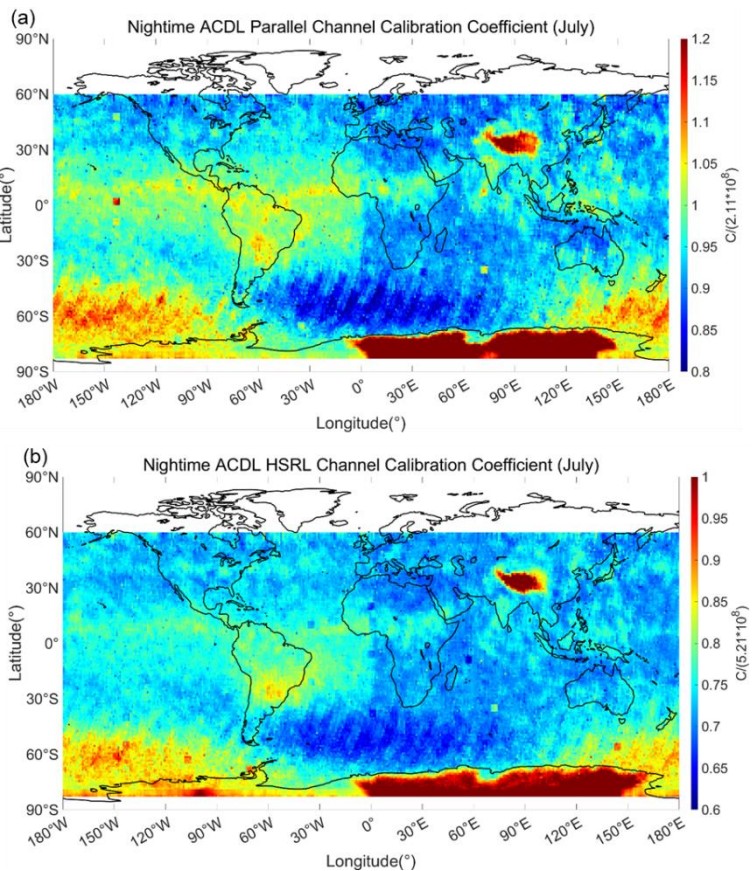

**Figure 12: Result of global calibration coefficient on a 1° latitude × 1° longitude grid for July, 2022. (a) The results of parallel channel and (b) the results of high-spectral-resolution channel.**



## 5. Summary

This paper presents a comprehensive calibration procedure for the first spaceborne high-spectral-resolution lidar with an iodine vapor absorption filter ACDL on board DQ-1 by utilizing nighttime 532 nm multi-channel data. The nighttime 532 nm multi-channel calibration procedure is established on the denoised raw data and combines the transmittance of scattered signals matched to elevation and geographic coordinates, to calculate the normalized signal. Extract valid profiles for calibration by using adaptive filters, thus obtain the daily calibration coefficients. The calibration coefficients after sliding averaging, are

used as the final results for calculating the global attenuated backscatter coefficient profiles. And the calibration coefficients for the cross-polarized channel relative to the parallel-polarized channel are determined through the utilization of the PGR.

This study analyzed the error sources of the multi-channel calibration coefficients and validated the results. The mean value of the attenuated backscatter coefficients in the calibration region shows a relative error of less than ~2%. The attenuated scattering coefficients validate that the ACDL polarization channel calibration is reliable and operates within the expected

error range of approximately 5%. The effective application of the ACDL nighttime calibration algorithm will enhance the calibration of the daytime orbit and other channels, thereby improving the quality of subsequent data products.

As the core component of the entire ACDL calibration procedure, the scientific team is committed to improving the daytime 532 nm calibration algorithm and the 1064 nm calibration algorithm. However, the calibration process has yet to account for the impact of background noise and high calibration coefficients in specific regions. In future research, the scientific team

plans to improve the nighttime 532 nm multi-channel calibration algorithm and introduce additional validation tools.

## Data availability

The ACDL data we used in this paper are not available publicly at the time when the article was submitted. We are allowed to access the data through our participation as a part of ACDL scientific team. The ERA5 dataset are downloaded via the website: https://cds.climate.copernicus.eu/cdsapp#!/dataset/reanalysis-era5-pressure-levels?tab=form (last access: 11 October 2023).

The SAGE III dataset are downloaded via the website: https://asdc.larc.nasa.gov/project/SAGE%20III-ISS(last access: 11 October 2023).

## Author contributions

F. Meng, J. Tang and G. Dai conceived and designed the Nighttime multi-Channel Calibration Algorithms; F. Meng and G. Dai wrote the manuscript; W. Long, X. Song, K. Sun and J. Liu participated in the algorithm development and data analysis;

J. Tang, S. Wu and W. Chen provided the supervision and participated in the scientific discussion. All the co-authors reviewed and edited the manuscript.



**Competing interests**

The authors declare that they have no conflict of interest.

**Acknowledgments**

This study has been jointly supported by the National Natural Science Foundation of China (NSFC) under grant U2106210, the Laoshan Laboratory Science and Technology Innovation Projects under grant LSKJ202201202, National Key Research and Development Program of China (2022YFB3901705), Qingdao Future Industry Cultivation Special Emerging Industry Plan 22-3-4-xxgg-8-gx.

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
