# Peer review of "ACDL/DQ-1 Calibration Algorithms. Part I: Nighttime 532 nm Polarization and High-Spectral-Resolution Channel"

_EGUsphere, 2024_

## Referee Comment (RC2)

**Review of "ACDL/DQ-1 Calibration Algorithms. Part I: Nighttime 532 nm Polarization and High-Spectral-Resolution Channel" by Meng et al.**

This paper must eventually be published so that prospective ACDL data users can understand how the fundamental calibration coefficients are derived and fully appreciate the uncertainties involved. However, the current manuscript is, regrettably, deficient in several areas. The following topics must be addressed in any revision.

a)  Add more details about the satellite (e.g., orbit altitude and inclination) and the lidar configuration (e.g., maximum measurement altitude, background subtraction region, etc.)

b)  Clarity in mathematical notation is essential. But in this manuscript, the notation is frequently ambiguous (e.g., equations 4, 21, 22, 24, 25) and sometimes demonstrably incorrect. Equation 27, reproduced below, is an especially egregious example.

$$\Delta X = \sqrt{\frac{\Sigma\left((X(z_j,y_k)-\hat{X}(z_j,y_k))-\frac{X(z_j,y_k)-\hat{X}(z_j,y_k)}{n}\right)^2}{n}}.$$

The authors assert that "$\Delta X$ represent the standard deviation (Std) of the differences between the actual measured values and the theoretical values". But (assuming I've done the algebra correctly!), simplifying their equation leads to this expression:

$$\Delta X = \left(\frac{n-1}{n}\right)\sqrt{\left(\frac{1}{n}\right)\sum\left(X\left(z_j,y_k\right)-\hat{X}\left(z_j,y_k\right)\right)^2}.$$

This is clearly not a standard deviation.

The authors need to revisit their equations everywhere in the manuscript to be sure that they accurately communicate the intended information.

c)  For some reason, the authors do not describe their method for calibrating the perpendicular channel measurements. To do this, the authors need to explain the operating principles of the ACDL "insertable depolarizer" (see Figure 1) and describe their method for determining the polarization gain ratio. Is the ACDL PGR obtained as described in Powell et al., 2009? Or is some other method employed? How are PGR uncertainties estimated? Do these uncertainties account for possible crosstalk between the polarization channels (e.g., Papetta et al., 2024)?

d)  It is essential that the authors describe how they combine the total backscatter (i.e., particulate + molecular) measured in the ACDL parallel and perpendicular channels with the molecular-only backscatter measured in their HSRL channel to produce profiles of particulate scattering ratios (or particulate backscatter coefficients). In presenting this material, they should derive an error budget that allocates fractions of the derived scattering ratio (or backscatter coefficient) uncertainty to influences such as the aerosol loading in the calibration region (which presumably only affects the parallel channel calibration coefficient), polarization purity in the laser transmitter and crosstalk in the receiver optics, altitude-dependent changes in the Fabry-Pérot etalon and iodine vapor absorption filter, etc.

e) Figure 8 demonstrates that the authors' procedure to fit the ACDL measurements to a molecular model is successful. This, however, is not sufficient to demonstrate that the ACDL is well-calibrated.

For calibration validation, the authors should be making comparisons of their HSRL measurements to independent data sets derived from different sensors. In a perfect world, underflights of the DQ-1 satellite would be conducted by a well-characterized airborne HSRL; e.g., perhaps the system developed at Zhejiang University (Wang et al., 2020), or maybe one currently being flown by investigators from Europe (Esselborn et al., 2008; Bruneau et al., 2015) or NASA (Hair et al., 2008; Burton et al., 2018). A comparison of the particulate scattering ratios (or backscatter coefficients) in the measurement overlap region of the two instruments would provide rock-solid validation. If airborne campaigns are not feasible, comparisons to aerosol backscatter coefficient profiles measured by ground based HSRLs or Raman lidars would be the next best alternative. Given fairly close spatial matching between the DQ-1 ground track and the ground stations, data from the Asian Dust Network (https://www-lidar.nies.go.jp/AD-Net/) and/or EARLINET (https://www.earlinet.org/) could provide excellent sources of high quality independent measurements.

Based on their statements on lines 451–453, the authors appear to recognize the need for the more rigorous calibration validation described above. Since the ACDL data is not currently available to the global science community, perhaps the authors should consider delaying the publication of this paper until they can incorporate thorough validation studies.

In addition to these general comments, below I am attaching an annotated version of the authors' manuscript that includes a large number of comments, questions, and suggestions. These too should all be addressed in a future revision of the manuscript.

[revised manuscript text omitted]

---

## Author Comment (AC1)

**Responses to RC2:**

This paper must eventually be published so that prospective ACDL data users can understand how the fundamental calibration coefficients are derived and fully appreciate the uncertainties involved. However, the current manuscript is, regrettably, deficient in several areas. The following topics must be addressed in any revision.

In addition to these general comments, below I am attaching an annotated version of the authors' manuscript that includes a large number of comments, questions, and suggestions. These too should all be addressed in a future revision of the manuscript.

AR: Thanks for investing your valuable time in thoroughly reviewing our manuscript and providing us with constructive feedback and suggestions.

General comments:

1.  Add more details about the satellite (e.g., orbit altitude and inclination) and the lidar configuration (e.g., maximum measurement altitude, background subtraction region, etc.)

AR: Thanks for your advice. We have included parameters for orbit altitude and inclination, and optical filter line widths in Table 1.

**Table 1: Parameters of the ACDL instrument for calibration**

| Parameters | Value |
| --- | --- |
| Wavelength | 532. 245 nm; |
| Pulse Energy | ~130 mJ@532 nm; |
| Gain | 59.46@parallel; 53.4573@perpendicular; 32@ high-spectral-resolution |
| Lidar Off-Nadir Angle | 2° |
| Laser Repetition Frequency | 40Hz@532nm |
| **Optical Filter Line Widths** | <30pm |
| Vertical Resolution | 3 m@<7.5 km; 24 m (8 bin average) @>7.5 km |
| Horizontal Resolution | ~ 330 m |
| **Orbit Altitude** | 705 km |
| **Orbit Inclination** | 98° |
| **Polarization Purity for PBS** | 1000:1 |

We have also added information about other lidar configurations, such as the maximum measurement altitude and background subtraction region, in the following passages.

**Line 107:** The maximum measurement altitude of the ACDL was defined as the "time delay", which was the time between the emission of the odd pulse and the start of signal acquisition, the maximum altitude

fluctuates in the range of 36-40 km (Dai et al., 2024).

**Line 115:** Both the nighttime parallel and high-spectral-resolution channels of ACDL were calibrated using the molecular normalization calibration technique, began with the denoised original data. The minimum values of the segmented-averaged signal in the parallel-polarized channel and perpendicular-polarized channel of 532 nm are selected as background noise (Yorks et al., 2015; Pauly et al., 2019; Palm et al., 2022), while the signals at high altitude are removed as background noise in the high-spectral-resolution channel (Dai et al., 2024).

*Reference:*

*Dai, G., Wu, S., Long, W., Liu, J., Xie, Y., Sun, K., Meng, F., Song, X., Huang, Z., and Chen, W.: Aerosol and cloud data processing and optical property retrieval algorithms for the spaceborne ACDL/DQ-1, Atmos. Meas. Tech., 17, 1879–1890, https://doi.org/10.5194/amt-17-1879-2024, 2024.*

2. Clarity in mathematical notation is essential. But in this manuscript, the notation is frequently ambiguous (e.g., equations 4, 21, 22, 24, 25) and sometimes demonstrably incorrect. Equation 27, reproduced below, is an especially egregious example.

$$\Delta X = \sqrt{\frac{\sum\left((X(z_j, y_k) - \hat{X}(z_j, y_k)) - \frac{X(z_j, y_k) - \hat{X}(z_j, y_k)}{n}\right)^2}{n}}$$

The authors assert that "$\Delta X$ represent the standard deviation (Std) of the differences between the actual measured values and the theoretical values". But (assuming I've done the algebra correctly!), simplifying their equation leads to this expression:

$$\Delta X = \left(\frac{n-1}{n}\right)\sqrt{\left(\frac{1}{n}\right)\sum\left(X(z_j, y_k) - \hat{X}(z_j, y_k)\right)^2}$$

This is clearly not a standard deviation. The authors need to revisit their equations everywhere in the manuscript to be sure that they accurately communicate the intended information.

AR: We feel sorry for our carelessness. It is correct to state that "$\Delta X$ represents the standard deviation (Std) of the differences between the actual measured values and the theoretical values", but the manuscript misunderstanding applies an equation to describe it. We have corrected the equation as follows:

$$\delta(z_{j,k}) = X(z_{j,k}) - \hat{X}(z_{j,k}) \tag{27}$$

$$\Delta X = \sqrt{\frac{\Sigma(\delta(z_{j,k})-\bar{\delta})^2}{n}}.$$  (28)

Where $\delta$ is the difference between measured and theoretical values, the $\bar{\delta}$ is mean of the $\delta$.

The detailed revisions of the other equations were described in the part of technical comments.

3.  For some reason, the authors do not describe their method for calibrating the perpendicular channel measurements. To do this, the authors need to explain the operating principles of the ACDL "insertable depolarizer" (see Figure 1) and describe their method for determining the polarization gain ratio. Is the ACDL PGR obtained as described in Powell et al., 2009? Or is some other method employed? How are PGR uncertainties estimated? Do these uncertainties account for possible crosstalk between the polarization channels (e.g., Papetta et al., 2024)?

AR: Thank you for raising this issue. As the principal optical path of the perpendicular-polarized channel is identical to that of the parallel-polarized channel, the different factors include the gain setting and beam splitter transmittance. Consequently, in the measurement, it is assumed that the operational state of the perpendicular channel is essentially equivalent to that of the parallel channel following normalization calculation. And the calibration of the perpendicular-polarized channel can be obtained by the relationship between the calibration coefficient of the parallel channel and the polarization gain ratio, as it is described by equations (9) and (23). To address this issue, we have added the description above in line 243.

**Line 243:** As the principal optical path of the perpendicular-polarized channel is identical to that of the parallel-polarized channel, the differentiating factor is the the gain setting and beam splitter transmittance. Consequently, in the measurement, it is assumed that the operational state of the perpendicular channel is essentially equivalent to that of the parallel channel. And the calibration of the perpendicular-polarized channel can be obtained by the relationship between the calibration coefficient of the parallel-polarized channel and the polarization gain ratio.

As you have mentioned, the ACDL is equipped with an insertable depolarizer to calibrate the polarization gain ratio on-board. However, based on the statistical analysis of the measured signals over a long time, it can be concluded that the gain of the ACDL is consistent with that of the ground at this stage. Concurrently, it is not necessary to activate the insertable depolarizer on board. In this study, we have

employed the laboratory-calibrated gain for each channel and conducted an error analysis based on the laboratory calibration results.

Since the PBS used in the ACDL has a polarization purity of 1000:1, the crosstalk between the polarization channels could be neglected and was included in the current error (~1%).

In subsequent on-board calibrations, we will continue to monitor and evaluate the gain of each channel, and the insertable depolarizer will be carried out when it is determined that the gain has indeed changed. The aforementioned text has been revised at line 253 and 413.

**Line 253:** The ACDL system has undergone ground calibration, including depolarization calibration. The calibration module has been specially reserved in the spaceborne ACDL (as shown in figure 1). In this module, a calibration beam with a known polarization state is pre-set. This, in combination with the usage of a half-wave plate, can be used for the on-orbit polarization calibrations (Alvarez et al., 2006; Powell et al., 2009; Freudenthaler, 2016).

**Line 413:** The currently used ground-calibrated PGR has a measurement error of about 1%, and the estimation includes both random and systematic errors, taking into account crosstalk between channels.

*Reference:*

*Alvarez, J.M., Vaughan, M.A., Hostetler, C.A., Hunt, W.H., and Winker, D.M.: Calibration Technique for Polarization-Sensitive Lidars, J Atmos Ocean Tech, 23, 683-699, https://doi.org/10.1175/JTECH1872.1, 2006.*

*Freudenthaler, V.: About the effects of polarising optics on lidar signals and the Δ90-calibration, Atmos Meas Tech, 9, 4181-4255, 10.5194/amt-9-4181-2016, 2016.*

4. It is essential that the authors describe how they combine the total backscatter (i.e., particulate + molecular) measured in the ACDL parallel and perpendicular channels with the molecularonly backscatter measured in their HSRL channel to produce profiles of particulate scattering ratios (or particulate backscatter coefficients). In presenting this material, they should derive an error budget that allocates fractions of the derived scattering ratio (or backscatter coefficient) uncertainty to influences such as the aerosol loading in the calibration region (which presumably only affects the parallel channel calibration coefficient), polarization purity in the laser transmitter and crosstalk in the receiver optics, altitude-dependent changes in the Fabry-Pérot etalon and iodine vapor absorption

filter, etc.

AR: Thank you for pointing out this issue. The ACDL comprises a high-spectral-resolution channel and polarization channels at 532 nm, thus enabling the direct calculation of particle scattering ratios from data measured by two channels. However, as a manuscript for introducing and evaluating the calibration algorithms, we analyze and evaluate the calibration algorithms for the different channels independently. The particle scattering ratio, as a subsequent data product, is not included in the analysis and evaluation of this paper. The validation methods subscribed in this paper, only used the atmospheric model data with known errors and assumed molecular scattering ratios as conditions.

In order to enhance the inappropriate presentation, amendments have been revised to the following paragraphs:

**Line 433:** To independently assessed the performance of calibration procedure in each channel, the pure-molecule attenuated scattering ratios were calculated by using attenuated backscatter data from the ACDL polarization channel and the HSRL channel with theoretical model values.

5.  Figure 8 demonstrates that the authors' procedure to fit the ACDL measurements to a molecular model is successful. This, however, is not sufficient to demonstrate that the ACDL is well-calibrated. For calibration validation, the authors should be making comparisons of their HSRL measurements to independent data sets derived from different sensors. In a perfect world, underflights of the DQ-1 satellite would be conducted by a well-characterized airborne HSRL; e.g., perhaps the system developed at Zhejiang University (Wang et al., 2020), or maybe one currently being flown by investigators from Europe (Esselborn et al., 2008; Bruneau et al., 2015) or NASA (Hair et al., 2008; Burton et al., 2018). A comparison of the particulate scattering ratios (or backscatter coefficients) in the measurement overlap region of the two instruments would provide rock-solid validation. If airborne campaigns are not feasible, comparisons to aerosol backscatter coefficient profiles measured by ground based HSRLs or Raman lidars would be the next best alternative. Given fairly close spatial matching between the DQ-1 ground track and the ground stations, data from the Asian Dust Network (https://www-lidar.nies.go.jp/AD-Net/) and/or EARLINET (https://www.earlinet.org/) could provide excellent sources of high quality independent measurements.

    Based on their statements on lines 451–453, the authors appear to recognize the need for the more rigorous calibration validation described above. Since the ACDL data is not currently available to

the global science community, perhaps the authors should consider delaying the publication of this paper until they can incorporate thorough validation studies.

AR: We are grateful for the valuable advice and for providing the data source. The underflight experiments synchronized with the on-board ACDL are undoubtedly the most effective way to validate the effectiveness of the calibration algorithms. Since the ACDL has not yet carried out these activities, so the content has not been included in this manuscript. The ACDL scientific team is engaged in the targeted development of an airborne lidar that can be utilized for the synchronized experiments. Our CAL/VAL team is monitoring this activity, which will be published in a subsequent paper. Since we are not involved in the AD-Net and EARLINET programs you have mentioned, it is challenging to obtain profile data for either of these lidar networks. Our team is actively strengthening the collaboration through the Dragon project (between ACDL and EarthCARE). In the future we will cross-validate with EarthCARE.

Due to the difficulty of conducting underflight experiments, we have used Raman Lidar from Chinese "The Belt and Road" Lidar Network to carry out ACDL validation as a supplement. The validation of ACDL has conducted a profile comparison utilizing a dual-wavelength polarization Raman lidar and the CALIPSO satellite. The Total Attenuation Backscatter Coefficient (TABC) and the Volume Depolarization Ratio (VDR) have been validated, respectively, and the validation results are presented in the following table (Liu et al., 2024).

Summary of ACDL passes over the ground-based lidar sites.

| Observed time (yyyy.mm.dd, local time) | Overpassed site | Closest distance point | Closest distance (km) | Weather condition | TABC bias (%) | VDR bias (%) |
|---|---|---|---|---|---|---|
| 2022.06.10, 14:25 | Zhangye | 100.5° E, 38.9° N | 7.0 | Clear | $-3.93 \pm 13.62$ | $-16.28 \pm 35.79$ |
| 2022.07.24, 14:24 | Zhangye | 100.5° E, 38.9° N | 0.2 | Clear | $-14.49 \pm 12.04$ | $-5.73 \pm 30.49$ |
| 2022.06.29, 03:41 | Dunhuang | 94.6° E, 40.1° N | 17.3 | Dust | $-25.15 \pm 18.84$ | $-18.24 \pm 40.79$ |
| 2022.07.10, 03:16 | Zhangye | 100.5° E, 38.9° N | 28.6 | Dust | $4.25 \pm 32.01$ | $6.27 \pm 36.23$ |
| 2022.05.27, 03:17 | Zhangye | 100.5° E, 38.9° N | 13.0 | Cloudy | $-8.33 \pm 30.53$ | $-9.07 \pm 54.96$ |
| 2022.07.06, 14:48 | Dunhuang | 94.6° E, 40.1° N | 8.4 | Cloudy | $-4.68 \pm 23.59$ | $-6.12 \pm 41.57$ |

The aforementioned contents have been revised to the following paragraphs of the manuscript.

**Line 482:** The underflight experiments synchronized with the on-board ACDL not yet implemented, steady progress is being made. The validation of ACDL profiles have conducted a profile comparison utilizing a dual-wavelength polarization Raman lidar and the CALIPSO satellite. The Total Attenuation Backscatter Coefficient (TABC) and the Volume Depolarization Ratio (VDR) have been validated,

respectively (Liu et al., 2024). The validation results further demonstrate the accuracy of the calibration algorithm.

*Reference:*

*Liu, Q., Huang, Z., Liu, J., Chen, W., Dong, Q., Wu, S., Dai, G., Li, M., Li, W., Li, Z., Song, X., and Xie, Y.: Validation of initial observation from the first spaceborne high-spectral-resolution lidar with a ground-based lidar network, Atmos. Meas. Tech., 17, 1403–1417, https://doi.org/10.5194/amt-17-1403-2024, 2024.*

Technical comments:

1. Line 31: What is the orbit altitude and inclination?

AR: Thanks for the suggestion. The orbit altitude and inclination are added to Table 1 as Orbit Altitude and Inclination.

**Table 1: Parameters of the ACDL instrument for calibration**

| Parameters | Value |
|---|---|
| Wavelength | 532. 245 nm; |
| Pulse Energy | ~130 mJ@532 nm; |
| Gain | 59.46@parallel; 53.4573@perpendicular; 32@ high-spectral-resolution |
| Lidar Off-Nadir Angle | 2˚ |
| Laser Repetition Frequency | 40Hz@532nm |
| **Optical Filter Line Widths** | <30pm |
| Vertical Resolution | 3 m@<7.5 km; 24 m (8 bin average) @>7.5 km |
| Horizontal Resolution | ~ 330 m |
| **Orbit Altitude** | 705 km |
| **Orbit Inclination** | 98˚ |
| **Polarization Purity for PBS** | 1000:1 |

2. Line 43: in practice you use the molecular signal, not the particulate

AR: Thanks for the suggestion. As described in this paper, the molecular signal is utilized as the standard particulate matter in atmosphere due to the inherent difficulty in determining the backscatter from aerosol or other atmospheric constituents. After having standard particulate matter as the calibration source, we could facilitate the conversion of the lidar signal to the backscatter exactly. The objective of this sentence is to highlight that on-board calibration of lidar is intended to facilitate the conversion of raw lidar signals to particle backscatter in a realistic measurement environment, rather than solely to molecules.

3. Line 51: "attenuation" ➔ "attenuated"

AR: Thanks, revised.

4. Line 56: "Ressell" ➔ "Russell"

AR: Thanks, revised.

5. Line 61: The Russell paper predates LITE by 15 years and refers only to uplooking lidars, so it's hard to see how any recommendations it offers on calibration altitudes would be relevant to space-based instruments. in fact, the Russell et al. claim that "normalization of lidar data above 30 km is not usually recommended" runs directly counter to the choices made for the LITE and CALIOP calibration regions.

AR: Thanks for the suggestion. As you correctly observed, this paper is not an appropriate source for citation in this context. Consequently, we have removed it from this revision. With regard to the claim by Russell et al. that "normalization of lidar data above 30 km is not usually recommended" and the concomitant assertion that "because there is usually a lack of radiosonde density data there", it is reasonable to assume that molecular backscattering above 30 km is difficult to accurately characterize due to the limitations of the measurement conditions at that time. With the advancement of measurement techniques, atmospheric conditions above 30 km can be effectively derived through the use of atmospheric models, including but not limited to data sets such as GMAO (Global Modeling and Assimilation Office) utilized by NASA and ERA5 employed in this paper. For these reasons, atmospheric data above 30 km can be utilized in the calibration algorithms of spaceborne lidar.

**Line 61:** The calibration of CALIOP underwent four versions, in the first three versions the atmosphere was used as the calibration altitude at 30-34 km, consistent with LITE. (Reagan et al., 2002; Hostettler et al., 2006; Powell et al., 2009).

6. Line 70: The correct reference for the CATS calibration algorithms is Pauly et al., 2019; https://doi.org/10.5194/amt-12-6241-2019

AR: Thanks, revised. "Yorks et al., 2016" ➔ "Pauly et al., 2019"

*Reference:*

*Pauly, R. M., Yorks, J. E., Hlavka, D. L., McGill, M. J., Amiridis, V., Palm, S. P., Rodier, S. D., Vaughan,*

*M. A., Selmer, P. A., Kupchock, A. W., Baars, H., and Gialitaki, A.: Cloud-Aerosol Transport System (CATS) 1064 nm calibration and validation, Atmos. Meas. Tech., 12, 6241–6258, https://doi.org/10.5194/amt-12-6241-2019, 2019.*

7. I'm accustomed to seeing these components referred to as either (a) perpendicular and parallel polarizations or (b) cross-polarized and co-polarized.  mixing the two conventions is unusual.

AR: Thanks for your kind reminder. The "Cross-polarized" has been revised to "Perpendicular-polarized" in the full manuscript.

8. Line 88: Note English language usage recommendations here

AR: Thanks, revised. "The entire parallel-polarized signal passes through a beam splitter (BS), while a portion (70%) of the parallel–polarized signal passes through an iodine vapor absorption filter to block Mie scattering, thus constituting the high-spectral-resolution channel." ➔ "The entire parallel-polarized signal passes through a beam splitter (BS), a portion (70%) of the signal through an iodine vapor absorption filter to block Mie scattering, thus constituting the high-spectral-resolution channel."

9. Line 90: not relevant to polarization separation or HSRL implementation; suggest moving this sentence elsewhere in the manuscript

AR: Thanks, deleted

10. Line 93: Missing "The"

AR: Thanks, revised.

11. Line 97: "hierarchical" ➔ "vertical"

AR: Thanks, revised.

12. Line 109: "based on" ➔ "applied to"

AR: Thanks, revised.

13. insert reference to equations 2 & 3

AR: Thanks, relocated.

14. There are several other parameters that should be included in this table; e.g.,

    (1) maximum measurement altitude

    (2) optical filter line widths

    (3) transmission/polarization purity specifications for the polarizing beam splitter cube

AR: Thanks for your advice.

(1)  We have added the information of maximum measurement altitude at line 107;

**Line 107:** The maximum measurement altitude of the ACDL was defined as the "time delay", which was the time between the emission of the odd pulse and the start of signal acquisition, the maximum altitude fluctuates in the range of 36-40 km (Dai et al., 2024).

(2)  We have included parameters for optical filter line widths in Table 1.

(3)  We have included parameters for transmission/polarization purity specifications for the polarizing beam splitter cube in Table 1.

**Table 1: Parameters of the ACDL instrument for calibration**

| Parameters | Value |
|---|---|
| Wavelength | 532. 245 nm; |
| Pulse Energy | ~130 mJ@532 nm; |
| Gain | 59.46@parallel; 53.4573@perpendicular; 32@ high-spectral-resolution |
| Lidar Off-Nadir Angle | 2° |
| Laser Repetition Frequency | 40Hz@532nm |
| **Optical Filter Line Widths** | <30pm |
| Vertical Resolution | 3 m@<7.5 km; 24 m (8 bin average) @>7.5 km |
| Horizontal Resolution | ~ 330 m |
| **Orbit Altitude** | 705 km |
| **Orbit Inclination** | 98° |
| **Polarization Purity for PBS** | 1000:1 |

15. Table 1: Inconsistent with figure 4, which suggests a center line of just over 532.245 nm

AR: We feel sorry for our carelessness. "532.024 nm" ➜ "532.245 nm"

16. Table 1: Measured on board or assumed to be constant based on commanded settings?

AR: The setting of gain is based on the results of the constants determined in the laboratory. The text has been revised at lines 253.

**Line** 253**:** The ACDL system has undergone ground calibration, including depolarization calibration.

The calibration module has been specially reserved in the spaceborne ACDL (as shown in figure 1). In this module, a calibration beam with a known polarization state is pre-set. This, in combination with the usage of a half-wave plate, can be used for the on-orbit polarization calibrations (Alvarez et al., 2006; Powell et al., 2009; Freudenthaler, 2016).

17. Table 1: Why are the gains in the parallel and perpendicular channels different? What design criteria are being satisfied by this choice?

AR: Thank you for pointing out this issue. Due to the influence of transmittance, sensitivity, and PMT optoelectronic magnification of each channel, so the gain shows different values in each channel. The gain used in this manuscript is the result of strict systematic calibration in the laboratory.

18. Table 1: "vertical" ➜ "perpendicular"

AR: Thanks, revised.

19. Line 113: "were" ➜ "are"

AR: Thanks, revised.

20. Table 1: "consists" ➜ "consisting"

AR: Thanks, revised.

21. Line 118: Cabannes line only? or full Rayleigh? (She, 2001; https://doi.org/10.1364/AO.40.004875)

AR: Cabannes line only. The relevant note was given in line 200 of the manuscript.

**Line 200**: The bandwidth of the F-P narrowband filter used in the ACDL is less than 30 pm, so 0.00366 was chosen as the ratio of perpendicular to parallel backscatter as the central Cabannes line where the backscatter can be detected (She, 2001; Cairo et al., 1999).

*Reference:*

*She, C.Y.: Spectral structure of laser light scattering revisited: bandwidths of nonresonant scattering lidars, Appl Optics, 40, 4875–4884, doi: 10.1364/ao.40.004875, 2001.*

*Cairo, F., Di Donfrancesco, G., Adriani, A., Pulvirenti, L., and Fierli, F.: Comparison of various linear*

*depolarization parameters measured by lidar, Appl Optics, 21, 4425–4432, doi: 10.1364/AO.38.004425, 1999.*

22. Line 120: How often does the "calculations of the transmittance effects due to the Fabry-Pérot etalon (F-P etalon) and the iodine vapor absorption filter" happen? i.e., are these calculations done for each laser pulse? for each orbit? daily? etc.

AR: These calculations are done for each laser pulse, so we have built a large and highly accurate look-up table to calculate them quickly.

23. Line 122: When I first read about the "denoised lidar signal" I was looking for a reference to the paper that describes the denoising algorithm. Adding a reference to section 2.3 will be a big help to future readers.

AR: Thank you for pointing out this issue. The ACDL lacks the background signal monitor, which would enable the reception of signals at high altitude and a considerable depth underground. In order to obtain an accurate background signal for each profile, it is necessary to analyze the signal characteristics and design a unique background signal algorithm to accurately capture the background signal. Consequently, this manuscript does not include the relevant literature on background noise algorithms and their designs. The ACDL CAL/VAL team will provide a detailed description in an subsequent paper (Pauly et al., 2019; Palm et al., 2022).

**Line 113:** The minimum values of the segmented-averaged signal in the parallel-polarized channel and perpendicular-polarized channel of 532 nm are selected as background noise (Yorks et al., 2015; Pauly et al., 2019; Palm et al., 2022), while the signals at high altitude are removed as background noise in the high-spectral-resolution channel (Dai et al., 2024).

*Reference:*

*Pauly, R. M., Yorks, J. E., Hlavka, D. L., McGill, M. J., Amiridis, V., Palm, S. P., Rodier, S. D., Vaughan, M. A., Selmer, P. A., Kupchock, A. W., Baars, H., and Gialitaki, A.: Cloud-Aerosol Transport System (CATS) 1064 nm calibration and validation, Atmos. Meas. Tech., 12, 6241–6258, https://doi.org/10.5194/amt-12-6241-2019, 2019.*

*Palm, S. P., Yang, Y., Herzfeld, U., and Hancock., D.: Ice, Cloud, and Land Elevation Satellite (ICESat–2) Project Algorithm Theoretical Basis Document for the Atmosphere, Part I: Level 2 and 3 Data Products, Version 6, NASA Goddard Space Flight Center, Greenbelt, MD 20771, 119pp., available at:*

*https://icesat–2.gsfc.nasa.gov/sites/default/files/page_files/ICESat2_*
*ATL04_ATL09_ATBD_PartI_r006.pdf (last access: 12 October 2023), 2022.*

24. Line 125: How can "the signal quality evaluation and atmospheric aerosol distribution" be done before the signals are calibrated? Or is external information used to assess aerosol loading? Please explain.

AR: The determination of the calibration altitude necessitates an assessment of both the signal quality of the on-board lidar and the vertical distribution of aerosols. As the calibration altitude increases, the signal quality of the LIDAR deteriorates, necessitating the use of longer averaging distances. Conversely, the errors due to aerosols increase. Here, we utilize CALIPSO and SAGE III observations to determine the aerosol loading within the calibration region. In summary, the ACDL selects 31–35 km as calibration region.

*Reference:*

*Vernier, J.P., Pommereau, J.P., Garnier, A., Pelon, J., Larsen, N., Nielsen, J., Christensen, T., Cairo, F., Thomason, L.W., Leblanc, T., and McDermid, I.S.: Tropical stratospheric aerosol layer from CALIPSO lidar observations, Journal of 595 Geophysical Research: Atmospheres, 114, doi: 10.1029/2009JD011946, 2009.*

[Figure]

Altitude-latitude cross sections of CALIOP scattering ratio at selected periods since the beginning of the mission (Vernier et al., 2009).

[Figure]

Backscattering ratios derived from SAGE III extinction data at 31-35 km altitude

25. Line 128: How is the polarization gain ratio determined?

AR: We apologize for this missing of the manuscript. The determination of the polarization gain ratio in the manuscript is same as it for the CALIOP. The ratio of the two detection channel signals is called the Polarization Gain Ratio, and it is used to quantify the differences in the responsivity and gain of the two 532-nm detection channels (Powell et al., 2009; Dai et al., 2024).

**Line 253:** The calibration for the perpendicular-polarized channel requires the application of the polarization gain ratio PGR of the perpendicular-polarized channel to the parallel channel, which defined as the ratio of the two detection channel signals., The polarization gain ratio is used to quantify the differences in the responsivity and gain of the two 532-nm detection channels (Powell et al., 2009; Dai et al., 2024):

26. Line 140: A table describing the onboard averaging intervals would be extremely useful; e.g., something similar to table 2 Winker et al., 2009 (https://doi.org/10.1175/2009JTECHA1281.1) or table 3 in Hunt et al., 2009 (https://doi.org/10.1175/2009JTECHA1223.1)

AR: Thanks for your advice. The ACDL data are only simply averaged on board, with a vertical resolution of 24m above 7.5km altitude and 3m below, and a uniform horizontal resolution of 330m, which is not necessary when presented in a table, but we added the above in the manuscript on line 133.

**Line 133:** After the geolocation and altitude corrections, the polarization and high-spectral-resolution channel data are obtained with a vertical resolution of 3 m at lower altitudes (below ~7.5 km), 24 m at higher altitudes (above ~7.5 km) along with a horizontal resolution of about 0.33 km.

27. Equation (1): The equation of "r" seems unnecessarily complicated. range is a fundamental quantity that is readily computed from the photon time of flight from the instrument to the sample volume. in general usage, I'd think altitude would be derived as a function of range.

AR: Thank you for pointing out this issue. We expect the ACDL calibration algorithm to be presented as a complete process, where the equations show that we perform a distance correction on the lidar data before proceeding to more detailed processing of the signal.

28. Equation (2) & (3): using too many symbols that are too similar risks introducing unwanted confusion (i.e., p in kp for laser pulse index, P for the received signal, and P for the "polarization channels"). I highly recommend a change in the notation to mitigate or even eliminate this potential problem.

This doesn't make any sense. what are these "system constants"? what aspects of the instrument do they characterize? how are they determined and what are their (nominal ranges of) values?

I note that neither KP or KM appear anywhere else in this manuscript. So the next version of the manuscript must either (a) provide detailed and convincing answers to these questions or (b) simply eliminate these terms from the equations. by adopting option (b), the "system constants" will be absorbed into the calibration coefficients.

AR: We feel sorry for our carelessness. To avoid unnecessary misunderstanding, we have deleted $K^P$ and $K^M$ from the manuscript and revised P used to denote received signals to S.

Equation (2) & (3):

$$X^P(z, k_p) = \frac{r^2 S^P(z, k_p)}{E_0(k_p) G_A} = C^P(k_p) \beta^P(z, k_p) T^2(z, k_p) f_{F-P}(z, k_p) \text{, and} \tag{2}$$

$$X^M(z, k_p) = \frac{r^2 S^M(z, k_p)}{E_0(k_p) G_A} = C^M(k_p) \beta^M(z, k_p) T^2(z, k_p) f_{F-P}(z, k_p) f_I(z, k_p) \text{,} \tag{3}$$

29. Line 150: According to figure 1, 30% of E0 is directed to X for the parallel polarization channel and 70% is directed to X for the molecular (i.e., HSRL) channel. How/Where is this energy difference reconciled when computing scattering ratios? Is it simply considered to be part of the gain ratio between the two?

AR: Thank you for pointing out this issue. The ACDL emits a pulsed laser with an energy of about 130

mJ and a polarization ratio of 500:1. In the receiving system, according to the optical efficiency and sensitivity of different channels, after receiving the perpendicular-polarized signals, the 3:7 optical splitter is designed to collect the parallel-polarized signals and the molecular channel signals, respectively, so that the signal intensity from each channel are basically the same by applying different gains to each channel. The ACDL has set up a monitor to ensure that the laser energy of each pulse can be monitored, and the subsequent calculations are performed after energy and gain normalized signals from each channel. **After energy and gain normalization**, the intensity of the measured signals from each channel is further guaranteed to be uniform. The contents of the scattering ratio calculated from the polarization and molecular channel signals collected by the ACDL itself are not included in this manuscript.

30. Line 151: Since $f_{F-P}$ and $f_I$ are altitude-dependent, they should be denoted as $f_{F-P}(z)$ and $f_I(z)$.

AR: Thanks, revised.

Line 151: The transmittance of F-P etalon and iodine vapor absorption filter of height dependence on the atmospheric temperature and pressure, which are denoted by $f_{F-P}(z)$ and $f_I(z)$.

31. Equation (4): for an integral with respect to r', the integrand should be a clear and obvious function of r'.

AR: Thanks, revised. Equation (4):

$$T^2(z, k_p) = \exp\left[-2\int_0^r \sigma(r')dr'\right].$$ (4)

32. Equation (5): use consistent notation; i.e., either σ(z, kp) or σ(z(kp), kp)

AR: Thanks, revised. Equation (5):

$$\sigma(z, k_p) = \sigma_m(z, k_p) + \sigma_{O_3}(z, k_p) + \sigma_a(z, k_p),$$ (5)

33. Line 160: "with footnote $m, O_3, a$ are on behalf of molecular scattering, aerosol scattering, and ozone absorption" ➔ "with subscript $m, O_3, a$ represent, respectively, molecular scattering, ozone absorption, and aerosol scattering."

AR: Thanks, revised.

34. Equation (8)-(10): Presumably the divisor also includes the $f_{F-P}$ and $f_I$ terms, yes? avoid ambiguity by stating so explicitly in the equations.

AR: Yes, we feel sorry for our carelessness, revised. Equation (8)-(10):

$$\beta'_{\parallel}(z, k_p) = \frac{X^{\parallel}(z,k_p)}{C^{\parallel}(k_p)f_{F-P}} = \beta^{\parallel}(z, k_p)T^2(z, k_p) , \tag{8}$$

$$\beta'_{\perp}(z, k_p) = \frac{X^{\perp}(z,k_p)}{C^{\parallel}(k_p)PGR(k_p)f_{F-P}} = \beta^{\perp}(z, k_p)T^2(z, k_p) \text{ and} \tag{9}$$

$$\beta'_{M}(z, k_p) = \frac{X^{M}(z,k_p)}{C^{M}(k_p)f_{F-P}f_I} = \beta^{M}(z, k_p)T^2(z, k_p) . \tag{10}$$

35. Line 170: How is the PGR determined?

AR: We apologize for this missing of the manuscript. The determination of the polarization gain ratio in the manuscript is same as it for the CALIOP. The ratio of the two detection channel signals is called the Polarization Gain Ratio, and it is used to quantify the differences in the responsivity and gain of the two 532-nm detection channels (Powell et al., 2009).

**Line 253:** The calibration for the perpendicular-polarized channel requires the application of the polarization gain ratio PGR of the perpendicular-polarized channel to the parallel channel, which defined as the ratio of the two detection channel signals., The polarization gain ratio is used to quantify the differences in the responsivity and gain of the two 532-nm detection channels (Powell et al., 2009):

36. Line 196: Please explain how the SAGE III extinction data are converted to backscatter; e.g., did the authors use an assumed lidar ratio? if so, what was the value? or did they instead use a technique similar to Knepp et al., 2020 (https://doi.org/10.5194/amt-13-4261-2020)?

AR: Thank you for pointing out this issue. In calculating the backscatter, we used the method provided by Knepp et al. The calculated aerosol backscatter coefficients were not used in the actual calculations, only to assess the aerosol loading. We apologize for the missing references in the manuscript, which have been added in the manuscript.

**Line 196:** In Eq. (13), the total backscattering coefficient of the parallel channel is subdivided into molecular volume scattering and aerosol volume scattering, with $\hat{\beta}^{\parallel}_m(z_c)$ and $\hat{\beta}^{\parallel}_a(z_c)$ are the parallel component of the molecular and the aerosol volume backscatter coefficient (Knepp et al., 2020), respectively. The calculated aerosol scatting ratio **were not used in the actual calculations**, only to

assess the aerosol loading.

*Reference:*

*Knepp, T. N., Thomason, L., Roell, M., Damadeo, R., Leavor, K., Leblanc, T., Chouza, F., Khaykin, S., Godin-Beekmann, S., and Flittner, D.: Evaluation of a method for converting Stratospheric Aerosol and Gas Experiment (SAGE) extinction coefficients to backscatter coefficients for intercomparison with lidar observations, Atmos. Meas. Tech., 13, 4261–4276, https://doi.org/10.5194/amt-13-4261-2020, 2020.*

37. Line 186: This is a circumflex, not a superscript. most often it is referred to as "hat notation"; e.g., see https://en.wikipedia.org/wiki/Hat_notation.

AR: Thanks, revised. "superscript." ➔ "hat nation".

38. Line 205: The "ACDL has selected a widely used value of $8\pi/3$ for the lidar ratio to calculate molecular, as $Sm = (8\pi/3) \times kbw$. This value is commonly used in the lidar community (Collins and Russell, 1976). And $k_{bw}$=1.0401 defines the dispersion of the refractive index and the King correction factor of air at 532 nm (She 2001; Hostetler et al., 2006; Reagan et al., 2002)." is a confusing sentence that appears to propose two different values for the molecular extinction-to-backscatter ratio (AKA lidar ratio). for clarity, consider condensing this paragraph into this single sentence:

   "In Eq. 16, Sm = $(8\pi / 3) \times$ kbw is the molecular lidar ratio (also extinction-to-backscatter ratio), where kbw = 1.0401 is the King factor that accounts for molecular anisotropy (Bucholtz, 1995; She, 2001; Hostetler et al., 2006)."

AR: Thanks, revised.

39. Line 214: "Different from Eq. (4), the attenuation for the ACDL to the calibration altitude $z_c$ can be described with" ➔ "Different from Eq. (4), the modeled attenuation for the ACDL to the calibration altitude $z_c$ can be described with"

AR: Thanks, revised.

40. Line 217: The sentence "Where $\hat{\sigma}$ with the footnote $m, O_3, a$ represents the extinction coefficients

of molecular, ozone and aerosol" is redundant.  subscript definitions already given on line 152.

AR: Thanks, deleted.

41. Line 219: "Chappius" ➔ "Chappuis"

AR: Thanks, revised.

42. Line 220: Consider citing Palm et al., 2002 rather than Yorks et al., 2016; e.g., see equations 3.2.7

and 3.2.8 in Palm et al.

AR: Thanks, revised.

*Reference:*

*Palm, S., W. Hart, D. Hlavka, E. J. Welton, A. Mahesh, and J. Spinhirne, 2012: The Algorithm Theoretical*

*Basis Document for the GLAS Atmospheric Data Products, NASA/TM–2012-208641 / Vol 6, 136 pp.;*

*https://ntrs.nasa.gov/citations/20120016*

43. Equation (21) & (22): Incorrect symbols

AR: Thanks, revised. Equation (21) & (22):

$$\hat{f}_{F-P}(T,P) = \int \mathcal{R}_m(T,P,\nu')\hat{F}_{F-P}(\nu')d\nu' \text{ and} \tag{21}$$

$$\hat{f}_I(T,P) = \int \mathcal{R}_m(T,P,\nu')\hat{F}_I(\nu')d\nu'. \tag{22}$$

44. Line 237: Use 1109 or 1111 iodine absorption lines?

AR: We feel sorry for our carelessness. The iodine molecular absorption filter of ACDL use iodine

absorption line 1110 (Dong et al., 2018).

*Reference:*

*Dong, J., Liu, J., Bi, D., Ma, X., Zhu, X., and Chen, W.: Optimal iodine absorption line applied for*

*spaceborne high spectral resolution lidar, Appl Optics, 5413-5419, doi: 10.1364/AO.57.005413, 2018.*

45. How is the PGR estimated?

AR: We apologize for this missing of the manuscript. The determination of the polarization gain ratio in

the manuscript is same as it for the CALIOP. The ratio of the two detection channel signals is called the Polarization Gain Ratio, and it is used to quantify the differences in the responsivity and gain of the two 532-nm detection channels (Powell et al., 2009).

**Line 253:** The calibration for the perpendicular-polarized channel requires the application of the polarization gain ratio PGR of the perpendicular-polarized channel to the parallel-polarized channel, which defined as the ratio of the two detection channel signals., The polarization gain ratio is used to quantify the differences in the responsivity and gain of the two 532-nm detection channels (Powell et al., 2009):

46. Line 252: The citation should also include Powell et al., 2009

AR: Thanks, revised.

47. Line 262: What is the maximum altitude measured by ACDL? Should a significant volcanic eruption occur (e.g., Pinatubo scale), can the calibration region be moved higher in the atmosphere?

AR: Thanks for your advice. The maximum measurement altitude of the ACDL was defined as the "time delay", which was the time between the emission of the odd pulse and the start of signal acquisition, the maximum altitude fluctuates in the range of 36-40 km (Dai et al., 2024). Since We combine the signal quality and the effect of aerosols in order to select the highest value under plausible data, the selection of 31-35km as the calibration region is almost the upper limit.

*Reference:*

*Dai, G., Wu, S., Long, W., Liu, J., Xie, Y., Sun, K., Meng, F., Song, X., Huang, Z., and Chen, W.: Aerosol and cloud data processing and optical property retrieval algorithms for the spaceborne ACDL/DQ-1, Atmos. Meas. Tech., 17, 1879–1890, https://doi.org/10.5194/amt-17-1879-2024, 2024.*

48. Line 267: Isn't a denoising/outlier rejection scheme applied before this data averaging step? (see line 116)

AR: We apologize for this missing of the manuscript. In this step, the high-energy events in the signal are first eliminated, followed by short-range averaging as introduced in Equations (24) & (25). We have added in this paragraph:

**Line 267:** Firstly, eliminate the high-energy events from signals in the calibration region, and averaged the signal horizontally at 3.6 km intervals.

49. Line 267: Should "Eqs. (28) and (29)" be "Eqs. (24) and (25)"?

AR: Thanks, revised.

50. Equation (24): Should this be i = k-5 and i = k+5?

    Why index over i in the denominator but not in the numerator? Are you invoking an assumption of horizontal homogeneity over 25 x 0.33 = 8.25 km? If so, say so: this paper should specifically identify all assumptions made in the calibration procedure.

AR: We feel sorry for the misunderstandings caused by the equation. The description "i=" is missing from equation (24), making $y_i$ in the numerator difficult to understand. This equation express that every 11 profiles are averaged, which facilitates the calculation and partially reduces the effect of random errors. Equation (24) is revised as follows

$$C(y_k) = \frac{1}{j_{31km}-j_{35km}+1}\sum_{j=j_{31km}}^{j=j_{35km}}\frac{\frac{1}{11}\sum_{i=k-5}^{i=k+5}X(z_j,y_i)}{\hat{\beta}(z_j,y_k)\hat{R}(z_j,y_k)\hat{T}^2(z_j,y_k)\hat{f}_{F-P}(z_j,y_k)}, \tag{24}$$

51. Equation (24): There's nothing in the manuscript text that explains the horizontal averaging over 11 profiles. Please add the rationale for this horizontal averaging and explain what governs the choice of 11 profiles.

AR: We would like to calculate the true calibration coefficients of the profiles by Eq. (24), so that the theoretical values can be obtained by sliding averaging over a long distance. Keeping the spatial resolution below 5 km could effectively remove the effect of the underlying terrain on the signal while taking into account the signal quality. The averaging distance could be longer, but we assumed that about the averaging 11 profiles could satisfy the computation.

**Line 277:** Keeping the spatial resolution below 5 km could effectively remove the effect of the underlying terrain on the signal while taking into account the signal quality. Therefore, the averaging of 11 profiles is used here for computation.

52. Equation (25): Recursive usage? I sure hope not! to eliminate confusion, use different symbols for

different stages of the averaging process; e.g., $C_{3.6km}(yk)$ on the left hand side of equation 24 and the right hand side of equation 25, and $C_{150km}(yk)$ on the left hand side of equation 25.

AR: We feel sorry for the misunderstandings caused by the equation. Eq. 25 will use the results of Eq. 24 for subsequent calculations. Equation (25) is revised as follows:

$$\tilde{C}(y_k) = \frac{1}{139}\sum_{k-69}^{k+69} C(y_k) \tag{25}$$

53. Line 276: Confusing/contradictory. Since equation 25 generates quantities averaged over 500 km, the tilde over the C should only appear on the left side equation 25.

AR: We feel sorry for our carelessness. Equation (25) is revised as follows

$$\tilde{C}(y_k) = \frac{1}{139}\sum_{k-69}^{k+69} C(y_k) \tag{25}$$

**Line 275:** Where i and j are the index for horizontal and vertical sample in one profile, y and z are the horizontal distance and vertical distance along the track. The hat notation ~ denotes the parameters that are smoothed every 500 km along the track.

54. Line 280: What "mean distance" is this? Are the authors suggesting that correlation increases with averaging distance? please revise this sentence to clarify the meaning.

AR: Thanks for your advice. Here the averaging distance should be expressed as an average of the distances in the vertical direction, i.e., under single-profile conditions, the choice of a longer range of calibrations will help to reduce the effect of random noise. Nevertheless, the application of such a lengthy averaging will inevitably result in an amplified error of the molecular backscatter due to the variability of atmospheric conditions at varying altitudes (as illustrate in figure below). We have revised the sentences below:

**Line 280:** Under single-profile conditions, the correlation between the lidar received signal and the ERA5 atmospheric model increases significantly with increasing averaged vertical distance, as the choice of a longer range of calibrations helps to reduce the effect of random noise. However, increasing the vertical distance is also accompanied by a significant increase in the variation of molecular backscattering coefficients. Therefore, ACDL chooses 31-35 km as the calibration region.

55. Line 280: Is the design of ACDL susceptible to these high energy events? e.g., were shielded PMTs

used? how frequently (and where) are these events encounterd by ACDL? Some discussion of this

topic (e.g., as in Hunt et al., 2009) would make a very useful addition to this manuscript (especially

since the effects of high energy particles on lidar calibration were well documented by the CALIPSO

project).

AR: For sure, the PMTs mounted on the ACDL are of shielded design. Excluding above the SAA region,

only a small number of high-energy events were observed all over the world, which is similar to the

conclusion from the CALIPSO mission (Hunt et al., 2009). The high-energy events observed by ACDL

will be further analyzed in a follow-up study.

*Reference:*

*Hunt, W.H., Winker, D.M., Vaughan, M.A., Powell, K.A., Lucker, P.L., and Weimer, C.: CALIPSO Lidar*

*Description and Performance Assessment, J Atmos Ocean Tech, 26, 1214–1228, doi:*

*10.1175/2009JTECHA1223.1, 2009.*

56. Line 219: "is" ➔ "can be"

AR: Thanks, revised.

57. Line 296: This aspect of the calibration procedures needs much, MUCH more explanation.

What is the rationale for using a "sliding average of 500 km in the direction of adjacent-track

distances". How long does one have to wait to achieve this 500 km distance? (One month? What

is the orbit repeat cycle of the ACDL platform?) What degree of thermal drift occurs during this

time period? I'd think thermal stability of the system would be much better assured if, instead of

using the closest spatial match, the authors used adjacent orbit tracks closest in time to the current

track (i.e., as CALIOP does). But, since ACDL is calibrated lower in the atmosphere than CALIOP,

maybe horizontal variability in stratospheric aerosol loading governs the design of the averaging

scheme?

As I say, much more detailed explanation is needed here.

AR: Thanks for your advice. The ACDL has a revisit cycle of about 50 days, the 500 km adjacent-track

average for calibration monthly, for the purpose of further calibrate the calibration coefficients. ACDL

performs **monthly** global calibrations that account for a variety of thermodynamically induced monthly

variations, including signal noise, device consistency, and slight instrument misalignment. The horizontal resolution of ACDL is 0.33km, and the average of 500 km along the orbit contains 1500 profiles, which is basically sufficient for the computation of calibration coefficients after data filtering. Subsequent adjacent-track averaging will further improve the stability of the calibration coefficients and help to carry out the assessment of the volatility of the calibration coefficients.

We have added the sentences below:

**Line 296:** The horizontal resolution of ACDL is 0.33 km, and the average of 500 km along the orbit contains ~1500 profiles, which is basically sufficient for the computation of calibration coefficients after data filtering.

58. Figure 5a: Is this an ascending node or a descending node? (see previous comments about the need for more info about ACDL orbit parameters)

AR: Ascending node. The ACDL enters a descending orbit during the night, and the orbit parameters have been added to Table 1.

**Table 1: Parameters of the ACDL instrument for calibration**

| Parameters | Value |
|---|---|
| Wavelength | 532. 245 nm; |
| Pulse Energy | ~130 mJ@532 nm; |
| Gain | 59.46@parallel; 53.4573@perpendicular; 32@ high-spectral-resolution |
| Lidar Off-Nadir Angle | 2˚ |
| Laser Repetition Frequency | 40Hz@532nm |
| **Optical Filter Line Widths** | <30pm |
| Vertical Resolution | 3 m@<7.5 km; 24 m (8 bin average) @>7.5 km |
| Horizontal Resolution | ~ 330 m |
| **Orbit Altitude** | 705 km |
| **Orbit Inclination** | 98˚ |
| **Polarization Purity for PBS** | 1000:1 |

59. Figure 5b: What are the units of the calibration coefficients?

AR: We apologize for this missing of the figure. The calibration coefficients of parallel-polarized channel normalized by $2.11*10^8$ $m^3srJ^{-1}$, and $5.21*10^8$ $m^3srJ^{-1}$ for the HSRL channel.

We have added the subscript below:

**Line 303:** Figure 5: The calibration coefficients for each calibration region, the calibration coefficients of parallel-polarized channel normalized by $2.11*10^8$ $m^3srJ^{-1}$, and $5.21*10^8$ $m^3srJ^{-1}$ for the HSRL channel.

60. Figure 5: What's responsible for the sudden increase at ~58°N?

AR: Since the study is for nighttime calibration only, in the example data the ACDL shifts to daytime at ~58°N.

61. Line 310: Also see Kar et al., 2018, Which describes updates to the CALIOP data filtering strategy that were implemented since Lee et al., 2008 was published

AR: Thanks for your advice. We have added the citation in manuscript.

**Line 310:** The lidar data contains random signal spikes, which can significantly impact the calibration coefficients (Lee et al., 2008; Kar et al., 2018).

*Reference:*

*Kar, J., Vaughan, M.A., Lee, K.P., Tackett, J.L., Avery, M.A., Garnier, A., Getzewich, B.J., Hunt, W.H., Josset, D., Liu, Z., Lucker, P.L., Magill, B., Omar, A.H., Pelon, J., Rogers, R.R., Toth, T.D., Trepte, C.R., Vernier, J.P., Winker, D.M., and Young, S.A.: CALIPSO Lidar Calibration at 532 nm: Version 4 Nighttime Algorithm, Atmos Meas Tech, 11, 1459–1479, doi: 10.5194/amt–11–1459–2018, 2018.*

62. Line 317: The calculation of "theoretical X" can be a tricky business, since generating "theoretical" values of X requires the use of "theoretical" calibration coefficients. How is this accomplished within the computational framework being described here?

AR: Thanks for pointing the issue. The measured signals of the ACDL correlate well with the modeled results, especially after sliding averaging over long distances. Therefore, when calculating the theoretical values, we first perform a sliding average of the ACDL measurement signals over 2000 km and calculate the theoretical "X" based on the mean value of the calibration coefficients calculated for the whole track.

63. Line 321: Please provide some insights into

   (a) How these scaling factors are chosen (e.g., arbitrarily or according to some heuristic);

   (b) The nominal range of these scaling factors; and

   (c) What fraction of the total data volume is excluded both in high noise regions (e.g., the SAA) and in low noise regions (e.g., northern hemisphere mid-latitudes)

AR: Thanks for pointing these issues.

(a) We performed a statistical analysis of the signal intensity distribution of high energy signals (as shown in the figure below). Since the ACDL signal basically follows the **Poisson distribution**, with drastic changes in the low value region and smoothness in the high value region, different factors are selected in the high and low value regions to ensure the accuracy of the data. The intensity of signal distributions that are significantly higher (or lower) than the measured values are used as the basis for selecting the scaling factors;

(b) In this research, the scale factors are chosen as -0.5 and 1.5;

(c) In the case of orbit 9808, the number of profiles taking into account is approximately 6,600. By using the filter, the rejection proportion between 20°N ~ 40°N is ~1.5%, and the rejection proportion between 10°S ~ 30°S (SAA region) is ~10%.

[Figure]

Signal distribution and filtering scheme for 33 km (Orbit 9808)

64. Equation 27: Summation over j? Summation over k? Both? There's no way to know based on what's written. This equation needs to be rewritten with clear, unambiguous notation.

Also, check the quantity shown in the red box. while I assume this is supposed to represent the mean of the differences, that's not what's written here.

AR: We feel sorry for the misunderstandings caused by the equation. Equation (27) is revised as follows

$$\delta(z_{j,k}) = X(z_{j,k}) - \hat{X}(z_{j,k}) \tag{27}$$

$$\Delta X = \sqrt{\frac{\Sigma(\delta(z_{j,k}) - \bar{\delta})^2}{n}}. \tag{28}$$

Where $z_{j,k}$ is the location of the calibration profile, $n$ is the number of the averaged bins, $\delta$ is the difference between measured and theoretical values, the $\bar{\delta}$ is mean of the $\delta$, $\Delta X$ represents the standard deviation (Std) of the $\delta$.

65. Equation 27: Isn't $z_j$ the location of a single range bin within profile $y_k$? If so, why not simplify your notation by using $z_{j,k}$?

AR: Thanks for your advice, revised.

66. Line 329: "represent" ➔ "represents"

AR: Thanks, revised.

67. Line 335: Define your terms. What is $X_{valid}$? Say so explicitly. Do not make users guess. (What's obvious to the authors, who have lived and breathed this material for years, may not always be obvious to new readers who are not intimately familiar with lidar data analyses.)

AR: Thanks for pointing the issues. We have added the subscript about $X_{valid}$ in manuscript.

Line 335: Use the data filtered in the first step as valid data for the subsequent NSR calculation.

68. Line 339: I'm confused again. How is it that we jump from a discussion of single profile noise rejection to daily estimates of the calibration coefficient? (this brings me back to an earlier question about the ACDL orbit repeat cycle.)

AR: We feel sorry for the misunderstandings caused by the sentence. This sentence was deleted from the manuscript due to the misunderstanding it created in this section.

69. Figure 6: Please comment: Is the elevated NSR below ~40°S due in part to low molecular number densities in the austral winter?

AR: Yes, we suppose that this is partly influenced by the fact that the number densities of molecules is too low, leading to insufficient detection signal strength in this region, which elevates the value of the NSR.

70. Figure 6: Add notation to the plots and/or text in the caption to explain that the different colors represent the 11 consecutive profiles used in the calculation of equation 24.

AR: Thanks, revised.

**Line 344:** The colored lines in the figure indicate 11 consecutive observation profiles.

71. Line 345: "along the" ➔ "as a function of"

AR: Thanks, revised.

72. Line 346: What considerations led to the selection of these two different values?

AR: Because the parallel and HSRL channels, as two channels with independent measurements, have different optical path and circuit, resulting in inconsistent signal strength between the two channels.

73. Line 348: "the comparies" ➔ "comparisons"

AR: Thanks, revised.

74. Line 349: "exclude" ➔ "excludes"

AR: Thanks, revised.

75. Figure 7: Please explain why the minimum threshold is chosen to be so much tighter than the maximum threshold

AR: The selection of the scaling factor is based on the statistical analysis of the signal intensity distribution of the high-energy signals, Since the ACDL signal basically follows the **Poisson distribution** (as shown in the figure below), with drastic changes in the low value region and smoothness in the high value region, different factors are selected in the high and low value regions to ensure the accuracy of the data. Therefore, in this study, the scaling factors were chosen to be asymmetrical -0.5 and 1.5.

[Figure]

Signal distribution and filtering scheme for 33 km (Orbit 9808)

76. Figure 7: according to the annotations in the top left corner, panel (c) does not show parallel channel results, but is instead a duplicate of panel (d).

AR: We feel sorry for our carelessness. In our resubmitted manuscript, the figure is revised.

Figure 7:

[Figure]

**Figure 7: Schematic of the original signal and calibration coefficients after filtering (orbit 9808), the calibration coefficients for parallel-polarized channel normalized by 2.11\*10$^8$ m$^3$srJ$^{-1}$, and 5.21\*10$^8$ m$^3$srJ$^{-1}$ for the HSRL channel. (a) The average signal $X^{\parallel}$ as a function of the corresponding latitude for altitudes**

**between 31 and 35 km, 1 July 2022. Within the SAA, there is a significant variation in the original signal, as indicated by the orange lines. The adaptive filter defines the minimum and maximum values with dotted lines, and the blue lines show the signals after filter. And the green dotted lines indicate the range of thresholds; (b) The average signal $X^M$ as a function of the corresponding latitude for altitudes between 31 and 35 km, orbit 9808. The lines in Figure 7b have the same meaning as in Figure 7a; (c) The filtered (blue lines) and unfiltered (orange lines) calibration coefficients of the parallel channel. The black line plots the smoothed calibration coefficients; (d) The filtered (blue lines) and unfiltered (orange lines) calibration coefficients of the high-spectral-resolution channel. The black line plots the smoothed calibration coefficients.**

77. Line 362: English language construction: the mean of the profiles is not what's being filtered. instead, the population of candidate profiles is being filtered prior to computing the mean.

AR: Thanks, revised.

**Line 362:** In the third step, the population of candidate profiles is being filtered prior to computing the mean using threshold determined by Eqs. (26) - (28) above.

78. Line 364: Wrong equation numbers again. Equations 32 and 33 describe aspects of the error analysis, not the data filtering. I believe the correct numbers are 26, 27, and 28?

AR: We feel sorry for our carelessness. "Eqs. (32) and (33)" ➔ "Eqs. (26) – (28)".

79. Line 365: Once again, I have no idea what the authors mean by this statement. (see my previous comment on line 304.) the authors need to fully explain the circumstances that lead to "the nearest calibration coefficients [being] selected as the daily calibration coefficient estimates for this area". in doing so, say whether "nearest" refers to nearest in time or nearest in space.
In general I'd think that rejecting some profiles simply means that the mean is determined from a smaller sample size and hence has a somewhat lower confidence. though I suppose that there could be cases where profiles are rejected over some extended contiguous along track distance. (I would also hope that this would be an unusual event.)

AR: Thanks for pointing out the problem. We would like to obtain profiles with unique calibration coefficient (the daily calibration coefficient), so that the averaging calibration coefficients could be more accurate. However, during the filtering, certain profiles that do not meet the requirements are removed as a result. Therefore, when generating the calibration coefficient, we use "the nearest calibration coefficient" to fill in these missing profiles. The "the nearest calibration coefficient" (guaranteed nearest

in distance) is defined as the previous or the next valid calibration coefficient.

To avoid misunderstanding, we have made revised the manuscript:

**Line 376:** The filtering causes some profiles being rejected, resulting in the calibration coefficients are empty in some regions, so the nearest calibration coefficients (guaranteed closest in distance) are selected as the complement.

80. Line 368: Delete the "lies"

AR: Thanks, revised.

81. Line 369: "at" ➔ "in"

AR: Thanks, revised.

82. Line 373: A map showing frequency of rejected samples (e.g., similar to figure 16 in Hunt et al., 2009) would make a nice addition to this manuscript.

AR: Thanks for your advice. We have added the figure 8 for subscript the percentage of the rejected profiles.

[Figure]

Figure 8: The percentage of rejected profiles for ACDL nightime calibration (July). (a) Parallel-polarized channel, and (b) HSRL channel

83. Line 390: Suggest citing Powell et al., 2009 here. That paper provides the rationale for random vs. systematic partitioning of errors adopted in this section of the manuscript.

AR: Thanks, we revised the manuscript in Line 390.

**Line 390:** The systematic uncertainty component of the parallel channel (Powell et al., 2009) is given by

84. Line 395: "enhanced" ➔ "revised"

AR: Thanks, revised.

85. Table 2: It's not clear to me how the authors arrived at this number based on the Cisewski paper. table 1 in that work states that the one of the pre-launch requirement for SAGE III measurements of aerosol extinction coefficients is a precision of 5%. but SAGE III extinction precisions are not sufficient for estimating the backscatter uncertainties needed to estimate ACDL systematic calibration errors. assuming the conversion from extinction to backscatter is done via an assumed lidar ratio, the uncertainty in that assumption must be combined with the uncertainty in the SAGE III extinction measurements.

Reagan et al., 2002 suggest a value of 0.04 for this parameter. while I do not know the genesis of the Reagan estimate, it seems more defensible than the authors' use of 0.03 attributed to Cisewski et al., 2014.

AR: Since Reagan et al. did not indicate the source of their data, we utilize CALIPSO and SAGE III observations to determine the aerosol loading within the calibration region. Based on the results of the data calculations and the aerosol loading decrease rapidly with altitude in the upper atmosphere, we consider 3% to be an acceptable error. As shown in the figure below, the averaged aerosol backscattering ratio at 31-35 km remains below 1.03 all year round, the reference (Cisewski et al., 2014) only indicates the use of SAGE III data as one of the sources. Due to the misunderstanding to accurately characterize the sources of error, we have been removed the reference from the manuscript.

[Figure]

Backscattering ratios derived from SAGE III extinction data at 31-35 km altitude

86. Equation (33): Notation problems (again). Is the summation over j or k? Why are C and C_hat indexed differently?

AR: We feel sorry for our carelessness. The Eq 33 has revised below:

$$\Delta C = \sqrt{\frac{\sum(C(z_{j,k})-\hat{C}(z_{j,k}))^2}{n}}.$$ (33)

87. Line 407: Why not express $\Delta C$ as a function of measurement SNR?

AR: We assume that the SNR is a function of the height z. Both z and the SNR can be used as variables to evaluate the random error in calibration coefficient C. Therefore, we have used the z as a basis for evaluating the random error here.

88. Equation (34): Careless equation presentation; the numerator should be $\Delta PGR$.

to repeat a previous question, where is the derivation of the PGR documented? also, how are PGR errors estimated? Do these error estimates account for crosstalk between the channels? Much more detail is required on this topic.

AR: We feel sorry for our carelessness. The Eq 34 has revised below:

$$\left(\frac{\Delta C^{\perp}}{C^{\perp}}\right)^2 = \left(\frac{\Delta C^{\|}}{C^{\|}}\right)^2 + \left(\frac{\Delta PGR}{PGR}\right)^2.$$ (34)

We apologize for this missing of the manuscript. The determination of the polarization gain ratio in the manuscript is same as it for the CALIOP. The ratio of the two detection channel signals is called the Polarization Gain Ratio, and it is used to quantify the differences in the responsivity and gain of the two 532-nm detection channels (Powell et al., 2009).

The ACDL have employed the laboratory-calibrated gain coefficients for each channel and conducted an error analysis based on the laboratory calibration results. Since the PBS used in the ACDL has a polarization purity of 1000:1, the crosstalk between the polarization channels could be neglected and was included in the current error.

The aforementioned text has been revised at lines below:

**Line 249:** The calibration for the perpendicular-polarized channel requires the application of the polarization gain ratio PGR of the perpendicular to the parallel-polarized channel, which defined as the ratio of the two detection channel signals. The polarization gain ratio is used to quantify the differences

in the responsivity and gain of the two 532-nm detection channels (Powell et al., 2009):

**Line 253:** The ACDL system has undergone ground calibration, including depolarization calibration. The calibration module has been specially reserved in the spaceborne ACDL (as shown in figure 1). In this module, a calibration beam with a known polarization state is pre-set. This, in combination with the usage of a half-wave plate, can be used for the on-orbit polarization calibrations (Alvarez et al., 2006; Powell et al., 2009; Freudenthaler, 2016).

**Line 413:** The currently used ground-calibrated PGR has a measurement error of about 1%, and the estimation includes both random and systematic errors, taking into account crosstalk between channels.

*Reference:*

*Alvarez, J.M., Vaughan, M.A., Hostetler, C.A., Hunt, W.H., and Winker, D.M.: Calibration Technique for Polarization-Sensitive Lidars, J Atmos Ocean Tech, 23, 683-699, https://doi.org/10.1175/JTECH1872.1, 2006.*

*Freudenthaler, V.: About the effects of polarising optics on lidar signals and the Δ90-calibration, Atmos Meas Tech, 9, 4181-4255, 10.5194/amt-9-4181-2016, 2016.*

89. Equation (35): Use consistent notation. In equations 2 and 3, X was previously defined as the range squared correct, gain and energy normalized signal. The "calibrated attenuated backscatter coefficients", $\beta'(z)$, are defined in equations 8 and 9.

AR: Thanks for pointing out the problem. The Eq 35 has revised below:

$$\Delta\beta'(z_{31-35},\ l) = \frac{\beta'(z_{31-35},l) - \widehat{\beta'}(z_{31-35},l)}{\beta'(z_{31-35},l)} \times 100\%, \tag{35}$$

90. Line 432: Notational consistency question: Why δ rather?

AR: Thanks for pointing this issue. We have revised "δ" to "Δ".

91. Line 439: If the assumption that the 26-30 km region was "pure molecular" was valid, wouldn't that be a better choice for applying the molecular normalization technique? The SNR is certainly higher there than between 31-35 km.

But it's easily established that this region is NOT purely molecular. To do this one could use the

same SAGE III data that the authors use to characterize aerosol loading in the ACDL calibration region. (question: what is the quotient of the mean SAGE III aerosol extinction coefficient between 26-30 km divided by the mean between 31-35 km?) Alternatively, one might even use CALIPSO data, as is done in figure 2 in Pauly et al., 2019 (https://doi.org/10.5194/amt-12-6241-2019).

AR: Of course, aerosols exist at 26-30 km region, and it is inevitable that aerosol loading will rise as altitude decreases. However, as a region used for validation, it is not necessary to guarantee the presence of aerosols over a large spatial extent. Here we only intercept a region with weak aerosol loading within a short distance, similar to the **"clear-air"** region (8-12 km) described by Powell et al., 2009. In view of CALIPSO's continuous monitoring of the "clear-air" region, we chose 26-30 km as the "pure-molecule" region for validation (Kar et al.,2018).

The averaged aerosol extinction coefficient measured by SAGE III at 520 nm is $1.96*10^{-4}$ km$^{-1}$ at 26-30 km and $1.41*10^{-5}$ km$^{-1}$ at 31-35 km, and the ratio of them is about 14 (June 2022).

At the same time, we wanted to perform the calibration without introducing errors into the external data (e.g. ~5% measurement error for SAGE III). Quoting SAGE III or CALIPSO measurements clearly does not meet this expectation. Due to the uncertainty in the global distribution of aerosols, it is more reasonable to use aerosols as a source of error rather than a correction condition within the current 31-35 km calibration region.

*Reference:*

*Powell, K.A., Hostetler, C.A., Vaughan, M.A., Lee, K., Trepte, C.R., Rogers, R.R., Winker, D.M., Liu, Z., Kuehn, R.E., Hunt, W.H., and Young, S.A.: CALIPSO Lidar Calibration Algorithms. Part I: Nighttime 532–nm Parallel Channel and 532–nm Perpendicular Channel, J Atmos Ocean Tech, 26, 2015–2033, doi: 10.1175/2009JTECHA1242.1, 2009.*

*Kar, J., Vaughan, M.A., Lee, K.P., Tackett, J.L., Avery, M.A., Garnier, A., Getzewich, B.J., Hunt, W.H., Josset, D., Liu, Z., Lucker, P.L., Magill, B., Omar, A.H., Pelon, J., Rogers, R.R., Toth, T.D., Trepte, C.R., Vernier, J.P., Winker, D.M., and Young, S.A.: CALIPSO Lidar Calibration at 532 nm: Version 4 Nighttime Algorithm, Atmos Meas Tech, 11, 1459–1479, doi: 10.5194/amt–11–1459–2018, 2018.*

92. Line 442: This is a testable proposition (e.g., using SAGE or CALIPSO). And since comparable correlative data is available, there's no need to rely on the authors' assumptions about what the

scattering ratio "should be".

AR: What we want to subscript here is that the ACDL, as a quantitative instrument, is capable of accurately measuring aerosols within the systematic error (~5%), thus providing the reader with a basis for subsequent evaluation of ACDL observations (e.g. Fig. 11). Therefore, the assumption of aerosol scattering ratios in the upper atmosphere under clear air (e.g. pure molecular) conditions has its applicability.

To avoid misunderstanding, we have deleted the "should be" and revised the sentence as follows:

**Line 441:** At low aerosol contents, the difference between the pure-molecule scattering ratios calculated from the calibrated attenuated backscatter coefficients and the molecular backscatter estimate is less than the systematic error (~5% for polarization channel and ~4% for HSRL channel), which is the relative calibration uncertainty.

93. Equation (36): Why? Since ACDL is an HSRL, there should be no need to resort to using attenuated scattering ratios. Instead, unattenuated aerosol scattering ratios are readily derived from equations 8, 9, and 10 given in section 2.1 of this manuscript.

AR: Yes, the ACDL does have the ability to measure the unattenuated aerosol scattering ratio. However, as a calibration work, we have to determine the measurement capability of each individual channel of the ACDL. If joint measurements from two different channels were used, the calculated aerosol scattering ratio results would undoubtedly superimpose the errors from both channels. In order to provide an accurate measurement error for each channel to the subsequent inversion (e.g. unattenuated aerosol scattering ratio), the unattenuated aerosol scattering ratios were not used here as validation.

94. Equation (37): Using aerosol scattering ratios eliminates any uncertainties involved in quantifying the aerosol two-way transmittance term

AR: Thanks.

95. Figure 11: I find it difficult to draw any useful insights or conclusions from these plots. More averaging would be a huge help.

AR: Thank you for your suggestions. As a graphical demonstration of the ACDL's capability for **single** profile measurement, Figure 11 shows the level of random error and the mean of the measurements for

the ACDL, with individual examples affected by random noise and aerosol, demonstrating different levels of aerosol scattering ratio.

96. Discussion and outlook: This section needs extensive revisions. The explanations offered by the authors for various artifacts in the spatial distribution of calibration coefficients are not plausible.

AR: Thanks for your kind reminder. We have revised the section below:

**Line 491:** Figure 13 illustrates the result of global calibration coefficient on a 1° latitude × 1° longitude grid for July, 2022. The Arctic and adjacent regions are in the polar day range in July, so there are no calibration coefficients for nighttime. During nighttime measurements, the strong backscatter targets (such as Tibetan Plateau and Antarctica) produced higher calibration coefficients than expected, which is due to the anomalously low background signal. When an ice target is detected, the background signal acquisition algorithm will extract low values from the subsurface as background values. Due to the lack of reliable dark noise detectors, the problem that occur currently will be revised by enhance the algorithms for background acquisition. The calibration procedures use denoised lidar signals and filters have reduced the effects of high-energy events, but strong signals due to low dark noise can still lead to the failures of calibration in these areas. In addition, elliptical regions of increased calibration coefficients near the poles can be seen, due to auroras near the poles (Hunt et al. 2009). The influence of these events also spreads as the sliding average frame progresses. Evaluate the impacts of different dark noise feature for lidar signals at night and rectify calibration coefficients in such regions will be conducted in the follow-up study.

97. Line 491: "illustrated" ➔ "illustrates"

AR: Thanks, revised.

98. Line 494: This phenomena wants for more and better explanation. The strong (presumably surface?) backscattering over the Tibetan Plateau and the Antarctic ice sheets should have no effect on nighttime signal magnitudes in the 31-35 km calibration region (during the daytime, the noise would be substantially larger but the mean backscatter signal should be comparable to the nighttime measurements).

AR: Thank you for pointing out this problem. Due to the strong correlation between regions with high

values of calibration coefficients and feature types, although we had doubts about feature types not affecting the data in high-elevation regions, we only assumed that it was the filters that were less effective in particular regions. As the study progressed, we realized that the background algorithm employed has some defects. The background noise exhibits lower signal levels in the ice region (figure below). Other groups in our team are working on and optimizing the background removal algorithms, please understand that due to the complexity of the algorithms that are common worldwide, we need to optimize and refine them further. In this manuscript we focus only on the calibration algorithm.

[Figure]

Background signal statistics (orbit 9928). Red circles indicate the ACDL passage over the Tibetan Plateau and Antarctic regions.

99. Line 493: (1) The increases in dark noise described in Hunt et al., 2009 are a characteristic of the CALIOP detectors. Are the authors suggesting that the ACDL detectors share the exact same characteristics?

(2) One would expect that increases in dark noise would increase the uncertainties in the calibration coefficients. But, assuming the data filtering scheme is working as expected, it's not immediately obvious how an increase in random noise in the signal would introduce a high bias in the calibration. However, the authors may want to review section 2.4 and (especially!) the material referencing figure 5 in Kar et al., 2018.  the molecular number densities over Antarctica during austral winter are typically quite low, and hence may require compensating adjustments to any NSR-based spike filter applied to profiles measured in that region and time.

AR: Thank you for pointing the problem.

(1) The ACDL background acquisition did observe high background in the SAA region similar to that observed by CALIOP in Hunt et al., 2009, but lacked the auroral portion of the acquisition;

(2) The issue of molecular number densities over Antarctica during the austral winter is indeed worth discussing; we will address the NSR spike-based filters in a future algorithm update.

100.Line 496: Don't nighttime background measurements provide an excellent proxy for dark noise?

AR: We have used the signal measured at night as the background signal and as dark noise. However, since the collected nighttime background is not high enough (or low enough), we assume that there are limitations to the algorithm currently. We will discuss the background noise algorithm and its implications for the calibration algorithm in more detail in a subsequent paper.

101.Line 500: "Evaluate" ➔ "Evaluation of"

AR: Thanks, revised.

102.Line 501: "rectify" ➔ "rectifying the"

AR: Thanks, revised.

103.Figure 13: (1) What causes the abrupt calibration discontinuity seen in the red ovals?

(2) Why are the calibration coefficients in the yellow ovals so much lower than elsewhere?

(3) Drawing the SAA boundaries on these plots would help readers identify regions that are especially prone to high noise excursions that might influence the magnitude of the calibration coefficients

AR: Thank you for pointing out these issues.

(1) As a schematic representation of the statistical results, the global results of the calibration coefficients have a number of problems that we have not noticed. The abrupt calibration discontinuity, as shown by the red ovals in the figure, is partly due to the visual illusion caused by the color bar. The other reason is due to errors in the global distribution and dispersion of the aerosols, and also the insufficient amount of data in this region.

(2) The lower molecular number densities at high latitudes in the Southern Hemisphere winter result in

a weaker signal measured by the ACDL, and thus lower calibration coefficients to increase the signal strength. We assume that the regions of high values of ellipsoid-like calibration coefficients on both sides of the same latitude are caused by auroras, and that the background removal algorithms are not effective in reducing the effects of auroras.

(3) Thanks for your advice, we added the circle that illustrates the SAA region to the figure 13.

[Figure]

Figure 13: Result of global calibration coefficient on a 1° latitude × 1° longitude grid for July, 2022. (a) The results of parallel channel and (b) the results of high-spectral-resolution channel. The red circles indicate the SAA region.

104. Reference: Attempting to retrieve a copy of this paper using this DOI gave me the following error message:

DOI Not Found: 10.5194/egusphere–2023–2182

This DOI cannot be found in the DOI System. The correct spelling is 10.5194/egusphere-2023-2182 (note the difference in the dashes used).

AR: We feel sorry for our carelessness. As the manuscript has been published, the reference has been revised as follow:

Reference:

Dai, G., Wu, S., Long, W., Liu, J., Xie, Y., Sun, K., Meng, F., Song, X., Huang, Z., and Chen, W.: Aerosol and cloud data processing and optical property retrieval algorithms for the spaceborne ACDL/DQ-1, Atmos. Meas. Tech., 17, 1879–1890, https://doi.org/10.5194/amt-17-1879-2024, 2024.

---

## Author Comment (AC2)

**Responses to RC1:**

This manuscript presents the calibration algorithms used for the ACDL lidar onboard the DQ-1 satellite, with the analysis of the 532 nm nighttime polarization and high-spectral-resolution channels. This work is of interest to the lidar community. Also, the upcoming launch of EarthCARE with the ATLID lidar in May 2024 will provide opportunity for further comparative studies between these two advanced lidar systems. I appreciate the efforts made in this study and offer some comments for improvement of this manuscript. Additionally, a careful review of the manuscript for English language improvements is advised, as some sections may benefit from further editing for clarity and grammatical accuracy.

AR: We greatly appreciate your valuable time for reviewing our research paper and providing feedback/suggestions.

General comments:

1. Figure 4: The blue dotted line is indistinguishable from other color curves representing iodine vapor absorption. Please enhance the figure's visualization quality for better interpretation.

AR: We feel sorry for our carelessness. The caption below Figure 4 has been corrected to show the **green dotted line** instead of the previously incorrect **blue dotted line**.

2. Figure 7 (a): The large variation in original signals within the SAA region is noticeable, but the overlay of filtered signals obscures the original data in most other regions. Adjusting the transparency level of the filtered data may resolve this issue. This recommendation applies to all subplots in Figure 7.

AR: Thanks for the suggestion. The figure 7 has been reprocessed by changing the line color of the original signal to orange and bold, adjusting color of the filtered data to blue and change the transparency.

[Figure]

Figure 7: Schematic of the original signal and calibration coefficients after filtering (orbit 9808). (a) The average signal $X^{\parallel}$ as a function of the corresponding latitude for altitudes between 31 and 35 km, 1 July 2022. Within the SAA, there is a significant variation in the original signal, as indicated by the orange lines. The adaptive filter defines the minimum and maximum values with dotted lines, and the blue lines show the signals after filter. And the green dotted lines indicate the range of thresholds; (b) The average signal $X^{M}$ as a function of the corresponding latitude for altitudes between 31 and 35 km, orbit 9808. The lines in Figure 7b have the same meaning as in Figure 7a; (c) The filtered (blue lines) and unfiltered (orange lines) calibration coefficients of the parallel channel. The black line plots the smoothed calibration coefficients; (d) The filtered (blue lines) and unfiltered (orange lines) calibration coefficients of the high-spectral-resolution channel. The black line plots the smoothed calibration coefficients.

3.  Figure 7 (c): There appears to be a discrepancy in the upper left legend; it likely should be labeled as "Parallel Channel" rather than "HSRL channel."

AR: We feel sorry for our carelessness. In our resubmitted manuscript, the figure is revised.

4.  The error analysis in this manuscript provides a foundation for assessing calibration accuracy. Can the authors discuss how these error metrics compare with those from other missions or standards in atmospheric lidar measurements, which can offer readers a benchmark for assessing ACDL's performance?

AR: Thanks for the suggestion. We focus on the CALIPSO mission, which is also a spaceborne lidar system for cloud and aerosol measurements, as the basis for our error analysis. The calibration procedure used by CALIPSO in version V3 selects a 30-34 km atmosphere as the calibration region, and provides

an error impact of 4% for aerosols in this region, combined with an error of 3% for purely molecular backscattering, which leads to a systematic error for calibration algorithm of ~5%. On this basis, CALIPSO has updated the calibration algorithm for latest version V4, which increases the calibration altitude to 36-39 km, which estimated to be 1.6±2.4% (Kar et al., 2018). And the systematic error of ACDL is higher than the calibration algorithm of CALIPSO night V4 version.

We add the text to Line 408 in the manuscript:

Line **408:** The error analysis of the ACDL nighttime calibration procedure refer to the CALIPSO mission, which is also a spaceborne lidar system for cloud and aerosol measurements. The V3 calibration procedure used by CALIPSO provides an error impact of 4% for aerosols, combined with the error of 3% for purely molecular backscatter, which leads to a systematic error for ~5% (Powell et al., 2009). And CALIPSO has updated the calibration algorithm for version V4, which increases the calibration altitude to 36-39 km, which the error estimated to be 1.6±2.4% (Kar et al., 2018). And the systematic error of ACDL is higher than the calibration algorithm of CALIPSO night V4 version

5. The manuscript mentions that ACDL data used in the paper are not publicly available. Given the scientific community's growing emphasis on open data for reproducibility and further analysis, consider discussing plans for data availability or establishing a data repository with access protocols, potentially making a portion of the data available.

AR: Thanks for the advice. We, as the CAL/VAL and Science Application Team, only consider the calibration and validation algorithms from a science application perspective. We're sorry that we can't decide when the data will be released, but we are actively promoting the sharing of the ACDL data. As far as we know, now there is relevant cooperation between ESA and CNSA (China National Space Administration) to promote data sharing mechanisms, and our team is also actively strengthening the cooperation through the Dragon project (between ACDL and EarthCARE). In short, we hope that more people will be able to use ACDL data.

6. The authors mentioned the plans for further validation tests. Can the authors expand on these plans, perhaps by detailing the types of lidar or other atmospheric sensors against which ACDL's data will be validated?

AR: Thanks for your kind reminder. The underflight experiments synchronized with the on-board ACDL are undoubtedly the most effective way to validate the effectiveness of the calibration algorithms. Since the ACDL has not yet carried out these activities, so the content has not been included in this manuscript. The ACDL scientific team is engaged in the targeted development of an airborne lidar that can be utilized for the synchronized experiments. Our CAL/VAL team is monitoring this activity, which will be published in a subsequent paper.

Due to the difficulty of conducting underflight experiments, we have used Raman Lidar from Chinese "The Belt and Road" Lidar Network to carry out the ACDL validation as a supplement. The validation of ACDL has conducted a profile comparison utilizing a dual-wavelength polarization Raman lidar and the CALIPSO satellite. The Total Attenuation Backscatter Coefficient (TABC) and the Volume Depolarization Ratio (VDR) have been validated, respectively, and the validation results are presented in the following table (Liu et al., 2024).

Table 1. Summary of ACDL passes over the ground-based lidar sites.

| Observed time (yyyy.mm.dd, local time) | Overpassed site | Closest point | Closest distance (km) | Weather condition | TABC bias (%) | VDR bias (%) |
|---|---|---|---|---|---|---|
| 2022.06.10, 14:25 | Zhangye | 100.5° E, 38.9° N | 7.0 | Clear | $-3.93 \pm 13.62$ | $-16.28 \pm 35.79$ |
| 2022.07.24, 14:24 | Zhangye | 100.5° E, 38.9° N | 0.2 | Clear | $-14.49 \pm 12.04$ | $-5.73 \pm 30.49$ |
| 2022.06.29, 03:41 | Dunhuang | 94.6° E, 40.1° N | 17.3 | Dust | $-25.15 \pm 18.84$ | $-18.24 \pm 40.79$ |
| 2022.07.10, 03:16 | Zhangye | 100.5° E, 38.9° N | 28.6 | Dust | $4.25 \pm 32.01$ | $6.27 \pm 36.23$ |
| 2022.05.27, 03:17 | Zhangye | 100.5° E, 38.9° N | 13.0 | Cloudy | $-8.33 \pm 30.53$ | $-9.07 \pm 54.96$ |
| 2022.07.06, 14:48 | Dunhuang | 94.6° E, 40.1° N | 8.4 | Cloudy | $-4.68 \pm 23.59$ | $-6.12 \pm 41.57$ |

The aforementioned contents have been revised to the following paragraphs of the manuscript.

**Line 482:** The underflight experiments synchronized with the on-board ACDL not yet implemented, steady progress is being made. As a supplement, the validation of ACDL profiles have conducted a profile comparison utilizing a dual-wavelength polarization Raman lidar and the CALIPSO satellite. The Total Attenuation Backscatter Coefficient (TABC) and the Volume Depolarization Ratio (VDR) have been validated, respectively (Liu et al., 2024). The validation results further demonstrate the accuracy of the calibration algorithm。

*Reference:*

*Liu, Q., Huang, Z., Liu, J., Chen, W., Dong, Q., Wu, S., Dai, G., Li, M., Li, W., Li, Z., Song, X., and Xie, Y.: Validation of initial observation from the first spaceborne high-spectral-resolution lidar with a*

*ground-based lidar network, Atmos. Meas. Tech., 17, 1403–1417, https://doi.org/10.5194/amt-17-1403-2024, 2024.*

Technical comments:

Lines 75-76: Simplify "with selected chose the region…" to "chose the region…" to enhance clarity.

AR: Thanks, revised.

Lines 80-82: Include the name of the lidar instrument onboard EarthCARE: the ATLID (Atmospheric Lidar).

AR: Thanks, revised.

**Line 81**: The upcoming deployment of the ATmospheric LIDAR (Light Detection and Ranging) lidar system, ATLID, is part of the payload of Earth Cloud, Aerosol and Radiation Explorer (Earth-CARE) mission has nearly finished ground-based calibration and performance verifications, with post-launch on-orbit calibrations scheduled to follow (Wehr et al., 2023).

Line 139: Suggest rephrasing for clarity: "Defining the normalized signals is a necessary first step for the different channels including:"

AR: Thanks, revised.

**Line 146:** Defining the range-scaled energy and gain-normalized signals (hereinafter normalized signal) is a necessary first step for the different channels including:

Line 151: Align the order of terms (molecular, ozone, aerosol) with their presentation in Eq(5).

AR: Thanks, revised.

Line 170: selects 31 -35 km as "the" calibration regions.

AR: Thanks, revised.

Line 171: Clarification needed – replace "subsection" with "the following paragraphs" if subsections are not explicitly defined.

AR: Thanks, revised.

Line 182: This sentence is unclear, please rephrase.

AR: Thanks, we have revised the sentence as follow:

**Line 190:** The ERA5 dataset provides hourly averaged global atmospheric parameters with 37 barometric pressure levels, and on a 0.25° latitude × 0.25° longitude resolution grid. The ERA5 global data is aligned with the altitude, latitude and longitude of the ACDL profiles.

The aerosol scattering ratio in Eq. 11 is given by the following equation

Line 197: If available, include more recent references regarding this lidar ratio selection.

AR: Thanks, we have added reference to line 205: Richard B Miles et al., 2001

Line 208: Ensure consistency in the order of terms as previously mentioned.

AR: Thanks, revised.

Line 285: Correct to "Eq. (26) is" since only one equation is referenced.

AR: Thanks, revised.

Line 289: Please provide details on defining the empirical scaling factor km, including the specific factor used in this study.

AR: Thanks for pointing these issues. We performed a statistical analysis of the signal intensity distribution of signals (as shown in the figure below). The intensity of signal distributions that are significantly higher (or lower) than the measured values are used as the basis for selecting the scaling factors. In this research, the scale factors are chosen as -0.5 and 1.5.

[Figure]

Signal distribution and filtering scheme for 33 km (Orbit 9808)

**Line 321:** The selection of the scaling factor is based on the statistical analysis of the signal intensity distribution of the high-energy signals, Since the ACDL signal basically follows the Poisson distribution, with drastic changes in the low value region and smoothness in the high value region, different factors are selected in the high and low value regions to ensure the accuracy of the data. Therefore, in this study, the scaling factors were chosen to be asymmetrical -0.5 and 1.5.

Figure 7: The subfigure labels (a)(b)(c)(d) are missing or unclear.

AR: Thanks, revised.

Line 328: Eqs. (32) and (33) are mentioned prematurely; they are introduced in the subsequent section.

AR: Thanks, revised. "Eqs. (32) and (33)" ➜ "Eqs. (26) - (28)".

Line 370: A delta symbol is missing before PGR; please verify.

AR: Thanks, revised.

Line 391: Amend to "can also be assessed" for grammatical accuracy.

AR: Thanks, revised.